



# Long-term validation of MIPAS ESA operational products using MIPAS-B measurements

Gerald Wetzel[1], Michael Höpfner[1], Hermann Oelhaf[1], Felix Friedl-Vallon[1], Anne Kleinert[1], Guido Maucher[1], Miriam Sinnhuber[1], Janna Abalichin[2], Angelika Dehn[3], and Piera Raspollini[4]

[1]Karlsruhe Institute of Technology, Institute of Meteorology and Climate Research, Karlsruhe, Germany

[2]Freie Universität Berlin, Institute of Meteorology, Berlin, Germany

[3]European Space Agency (ESA-ESRIN), Frascati, Italy

[4]Istituto di Fisica Applicata "N. Carrara" (IFAC) del Consiglio Nazionale delle Ricerche (CNR), Firenze, Italy

*Correspondence to:* Gerald Wetzel (gerald.wetzel@kit.edu)

## Abstract

The Michelson Interferometer for Passive Atmospheric Sounding (MIPAS) was a limb-viewing infrared Fourier transform spectrometer that operated from 2002 to 2012 aboard the Environmental Satellite (ENVISAT). The final re-processing of the full MIPAS mission Level

2 data was performed with the ESA operational version 8 (v8) processor. This MIPAS data set not only includes retrieval results of pressure-temperature and the standard species $H_2O$, $O_3$, $HNO_3$, $CH_4$, $N_2O$, and $NO_2$, but also vertical profiles of volume mixing ratios of the more difficult to retrieve molecules $N_2O_5$, $ClONO_2$, CFC-11, CFC-12 (included since v6 processing), HCFC-22, $CCl_4$, $CF_4$, $COF_2$, and HCN (included since v7 processing). Finally, vertical profiles

of the species $C_2H_2$, $C_2H_6$, $COCl_2$, OCS, $CH_3Cl$, and HDO were additionally retrieved by the v8 processor.

The balloon-borne limb-emission sounder MIPAS-B was a precursor of the MIPAS satellite instrument. Several flights with MIPAS-B have been carried out during the 10 years operational phase of ENVISAT at different latitudes and seasons, including both operational periods where

MIPAS measured with full spectral resolution (FR mode) and with optimized spectral resolution (OR mode). All MIPAS operational products (except HDO) were compared to results inferred from dedicated validation limb sequences of MIPAS-B. To enhance the statistics of vertical profile comparisons, a trajectory match method has been applied to search for MIPAS coincidences along 2-day forward/backward trajectories running from the MIPAS-B

measurement geolocations. This study gives an overview of the validation results based on the





ESA operational v8 data comprising the MIPAS FR and OR observation periods. This includes an assessment of the data agreement of both sensors taking into account combined errors of the instruments. The difference between retrieved temperature profiles of both MIPAS instruments generally stays within ±2 K in the stratosphere. For most gases, namely $H_2O$, $O_3$, $HNO_3$, $CH_4$,

$N_2O$, $NO_2$, $N_2O_5$, $ClONO_2$, CFC-11, CFC-12, HCFC-22, $CCl_4$, $CF_4$, $COF_2$, and HCN we find a 5 % to 20 % agreement of the retrieved vertical profiles of both MIPAS instruments in the lower stratosphere. For the species $C_2H_2$, $C_2H_6$, $COCl_2$, OCS, and $CH_3Cl$, however, larger differences within 20 % and 50 % appear in this altitude range.

## 40    1    Introduction

Satellite measurements of stratospheric trace gases are essential for monitoring the distribution and trend of these species on a global scale (Hegglin et al., 2021). The Environmental Satellite (ENVISAT) of the European Space Agency (ESA) operated ten years between 2002 and 2012. The Michelson Interferometer for Passive Atmospheric Sounding (MIPAS; Fischer et al., 2008)

was one of three chemistry instruments aboard ENVISAT, besides the Scanning Imaging Absorption Spectrometer for Atmospheric Chartography (SCIAMACHY; Bovensmann et al., 1999) and the Global Ozone Monitoring by Occultation of Stars (GOMOS) instrument (Bertaux et al., 1991). Validating instruments like MIPAS for the purpose of assessing measurement accuracy is an essential task. Stratospheric balloon measurements are particularly suitable to

reach this goal since these instruments are able to sound the atmosphere with high vertical resolution (e.g. Cortesi et al., 2007; Ridolfi et al., 2007; Wang et al., 2007; Wetzel et al., 2007; Payan et al., 2009; Wetzel et al., 2013a). The main logistical issue that the satellite and the validating balloon instruments observe the same air masses has to be considered carefully when performing balloon campaigns. Two principal comparison methods are common: (1) direct

matches where the balloon instrument measures at the same time and location where the satellite observation takes place and (2) trajectory matches where forward and backward trajectories are calculated from the balloon measurement geolocation to search for appropriate satellite overpasses. Several flights with the balloon version of MIPAS (Friedl-Vallon et al., 2004) were carried out during the operational time of ENVISAT. This study presents an overview of 10

years of MIPAS observations based on the recent ESA processor version 8 (v8; released in 2021; ESA, 2021; Dinelli et al., 2021) and provides the evaluation of the long-term performance of MIPAS covering the complete set of atmospheric parameters (except HDO) that have been



processed with the newest operational data version. In the following sections, validation activities, data analysis and validation results are described in detail.


## 2 Instruments and data analysis

### 2.1 MIPAS operations and data version

The limb-viewing Fourier transform spectrometer MIPAS on ENVISAT (hereinafter also referred to as MIPAS-E to better distinguish from the balloon instrument MIPAS-B) has been
designed to operate in the mid-infrared spectral region covering five spectral bands between 685 and 2410 $cm^{-1}$ with a maximum optical path difference (MOPD) of 20 cm, equivalent to an unapodized full spectral resolution of 0.025 $cm^{-1}$ (Fischer et al., 2008). The vertical instantaneous field of view (IFOV) was about 3 km. ENVISAT was launched into its sun-synchronous orbit by ESA on 1 March 2002 with 14.3 orbits/day and an Equator crossing local
solar time 10:00 (descending node). After the commissioning phase, MIPAS was run predominantly in its nominal measurement mode with full spectral resolution (called FR mode) from July 2002 until the end of March 2004. During each orbit approximately 72 limb scans covering tangent altitudes between 8 and 68 km were recorded (in steps of 3 km below 45 km) in the FR mode. The majority of validation studies based on correlative measurements
published so far were addressing MIPAS data recorded during this first time period. These measurements were originally reprocessed by the ESA Instrument Processing Facilities (IPF) v4.1 and v4.2, based on the Optimized Retrieval Model (ORM) code described in Ridolfi et al. (2000) and Raspollini et al. (2006), and covered pressure-temperature and the six constituents $O_3$, $H_2O$, $CH_4$, $N_2O$, $HNO_3$, and $NO_2$. The validation studies addressed these parameters and
constituents: pressure-temperature (Ridolfi et al., 2007), $O_3$ (Cortesi et al., 2007), $HNO_3$ (Wang et al., 2007), $NO_2$ (Wetzel et al., 2007), $N_2O$ and $CH_4$ (Payan et al., 2009), and $H_2O$ (Wetzel et al., 2013a).

After an increasing frequency of problems with the interferometer drive system in late 2003 and beginning of 2004 and upon subsequent detailed investigations it was decided to suspend
the nominal operations from March 2004 onwards for detailed investigations. From January 2005 onwards, the instrument was back to operation but at reduced MOPD (41 % of nominal) while maintaining the interferogram scan speed. During data processing, the interferograms were truncated to 8 cm MOPD, resulting in an unapodized spectral resolution of 0.0625 $cm^{-1}$.



The shorter acquisition time per interferogram led to the benefit of an equivalent improvement

in the vertical and horizontal (along-track) sampling. The duty cycle of this so-called optimized resolution (OR) mode (optimized in terms of a trade-off between spectral and spatial resolution considering instrument operation safety aspects) could be steadily increased from 30 % in January 2005 to 100 % from December 2007 on. MIPAS was successfully operated with this full duty cycle in the OR mode until 8 April 2012, when an ENVISAT anomaly occurred

resulting into the loss of communication between ground and satellite and the end of MIPAS observations (ESA, 2012). Details of the characteristics of the two MIPAS mission phases (FR and OR modes) in terms of instrument settings and atmospheric sampling are described in Raspollini et al. (2013).

The coarser spectral but finer spatial sampling of MIPAS since 2005 along with the need for

near real time analysis demanded adaptations in the calibration scheme and the processing codes. This was realized in ESA Level (L) 2 processor version (v) 6 and explained by Raspollini et al. (2013), which also provides the diagnostics of the products including the error budgets as estimated by Dudhia et al. (2002). The whole MIPAS data set covering almost 10 years of observations was re-processed with v6, v7, and v8. In addition, the number of retrieved

constituents was extended to $ClONO_2$, $N_2O_5$, CFC-11, and CFC-12 in ESA version 6.

ESA version 7 data (released in 2015) also includes the species HCN, HCFC-22, $CF_4$, $COF_2$, and $CCl_4$. The ESA L1v8/L2v8 diagnostic data set (DDS) version was released in June 2018 followed by the L1v8/L2v8 full mission (FM) data in June 2019. The new v8 data release (ESA, 2021; Dinelli et al., 2021; Raspollini et al., 2022) comprises the additional molecules $C_2H_2$,

$C_2H_6$, $COCl_2$, OCS, $CH_3Cl$, and HDO. For the final ESA reprocessing of MIPAS data, numerous improvements were implemented in the L2 processor Optimised Retrieval Model (ORM) version 8.22 (v8) and its auxiliary data including an update of the spectroscopic data used (Raspollini et al., 2022). All molecules except HDO have been validated by comparison with observations of the MIPAS balloon instrument.

**2.2   MIPAS-B data set**

The balloon-borne limb-emission sounder MIPAS-B can be regarded as a precursor of the MIPAS satellite instrument (Friedl-Vallon et al., 2004). Hence, a number of specifications like spectral resolution and spectral coverage are similar. The unapodized full spectral resolution is 0.0345 cm$^{-1}$, which is slightly coarser than the FR mode resolution but higher than the OR mode





resolution. However, for essential parameters the MIPAS-B performance is superior, in terms of NESR (Noise Equivalent Spectral Radiance) and line of sight (LOS) stabilization. The LOS is stabilized using an inertial navigation system supplemented with an additional star reference system which leads to an after all knowledge of the tangent altitude of better than 50 m at the $1\sigma$ confidence limit (Wetzel et al., 2010). The MIPAS-B NESR is further improved by

averaging multiple spectra recorded at the same elevation angle. The general data processing from interferograms to calibrated spectra including instrument characterization is described in Friedl-Vallon et al. (2004) and references therein.

    MIPAS-B measurements were recorded typically at a 1.5 km vertical tangent altitude grid. Retrieval calculations of temperature and atmospheric trace species were performed at a 1 km

grid with a Gauss-Newton iterative method (Rodgers, 2000) using analytical derivative spectra calculated by the Karlsruhe Optimized and Precise Radiative transfer Algorithm (KOPRA; Stiller et al., 2002; Höpfner et al., 2002). To avoid retrieval instabilities due to oversampling of vertical grid points, a regularization approach according to the method described by Tikhonov (1963) and Phillips (1962) constraining with respect to the first derivative of the a priori profile

was adopted. The resulting vertical resolution is typically between 2 and 5 km for the analysed atmospheric parameters and is therefore comparable or slightly better than the vertical resolution of the MIPAS satellite instrument. Table 1 gives an overview on the spectral windows used for the MIPAS-B target parameter retrievals. Different spectral microwindows within mostly the same molecular bands were used for the MIPAS-E data analysis (Dinelli et

al., 2021). Spectroscopic parameters for the calculation of the infrared emission spectra originate from the high-resolution transmission (HITRAN) molecular absorption database (Rothman et al., 2009) and a MIPAS dedicated spectroscopic database (Raspollini et al., 2022). For heavy molecules like CFC-11, CFC-12, HCFC-22, $CCl_4$, and $CF_4$, new and improved infrared absorption cross sections (Harrison, 2015; Harrison, 2016; Harrison et al., 2017) were

used for the calculation of radiative transfer (consistent to the MIPAS-E retrieval).

    The MIPAS-B error budget includes random noise as well as covariance effects of the fitted parameters, temperature errors, pointing inaccuracies, errors of non-simultaneously fitted interfering species, and spectroscopic data errors ($1\sigma$). For further details on the MIPAS-B data analysis and error estimation, see Wetzel et al. (2012; 2015) and references therein. An

overview of typical errors for the atmospheric parameter retrieval is given in Table 1.



## 2.3 Validation approach

A number of MIPAS balloon flights have been carried out as part of the validation program of the chemistry instruments aboard ENVISAT. Most of the MIPAS-B data used here, however, were obtained during flights that were done in the framework of various scientific projects.

MIPAS-B had a sophisticated pointing system so that the full freedom of a balloon-borne limb emission sounder in terms of observation time, viewing direction and sampling strategy could be used to get the best possible coincidence in time and space with the satellite overpass even during balloon flights that were not primarily dedicated to satellite validation. If compliant with the scientific goal of the mission and the weather conditions, the strategy was to launch the

balloon in due time before an ENVISAT overpass and to optimize the azimuthal viewing direction and the vertical sampling at the time of the overpass. Except for two flights, a coincidence in space and time between both sensors could be achieved such that vertical profiles of both instruments can be directly compared. An overview of the MIPAS balloon flights used in this study is given in Table 2.

To enhance the statistics of profile comparisons, diabatic 2-day forward and backward trajectories were calculated by the Free University of Berlin using a trajectory model (Naujokat and Grunow, 2003; Grunow, 2009). The trajectories are based on European Centre for Medium-Range Weather Forecasts (ECMWF) 1.25°x1.25° analyses and start at different altitudes at the geolocation of the balloon observation to search for a coincidence with the satellite

measurement along the trajectory path within a match radius of 1 h and 500 km. Temperature and volume mixing ratio (VMR) of the satellite match have been interpolated to the trajectory match altitude such that these values can be directly compared to the MIPAS-B data at the trajectory start point altitude. Altitude differences between the trajectory start and match point have to be taken into account in the case of temperature by means of an adiabatic correction.

The handling of the diurnal variation of photochemically active species is discussed below.

The primary vertical coordinate of MIPAS-E is pressure whereas for MIPAS-B it is altitude. For all intercomparisons shown in this study, vertical profiles refer to the MIPAS-B pressure-altitude grid. Differences between measured quantities of MIPAS-E and the validation instrument MIPAS-B are expressed in absolute and relative units. The mean difference $\Delta x_{mean}$

for $N$ profile pairs of compared observations is given as:

$$\Delta x_{mean} = \frac{1}{N} \sum_{n=1}^{N} (x_{E,n} - x_{B,n}) \,, \tag{1}$$

where $x_E$ and $x_B$ are data values of MIPAS-E and MIPAS-B at one altitude level respectively. The mean relative difference $\Delta x_{mean,rel}$ of a number of profile pairs is calculated by dividing the mean absolute difference by the mean profile value of the reference instrument MIPAS-B:

$$\Delta x_{mean,rel} = \frac{\Delta x_{mean}}{\frac{1}{N} \sum_{n=1}^{N} x_{B,n}} \cdot 100\% \;. \tag{2}$$

Differences are displayed together with the combined errors $\sigma_{comb}$ of both instruments, which are defined as:

$$\sigma_{comb} = \sqrt{\sigma_E^2 + \sigma_B^2} \,, \tag{3}$$

where $\sigma_E$ and $\sigma_B$ are the precision, systematic or total errors of MIPAS-E and MIPAS-B, 
respectively.

Precision errors characterize the reproducibility of a measurement and correspond, in general, to random noise errors. Systematic errors used for the MIPAS-E data analysis have been assessed in corresponding studies (Dudhia et al., 2002; Raspollini et al., 2013; Dinelli et al., 2021). The uncertainty of the calculated mean difference (standard error of the mean, SEM) is
given by $\sigma/N^{0.5}$ where $\sigma$ is the standard deviation (SD). A bias between both instruments is considered significant if the SEM is smaller than the bias itself. The comparison between the VMR difference and the combined systematic error (for statistical comparisons) or total error (for single comparisons) is appropriate to identify unexplained biases in the MIPAS-E measurements when they exceed these combined error limits. Since the vertical resolution of
the atmospheric parameter profiles of both instruments is of comparable magnitude, a smoothing by averaging kernels has not been applied to the observed profiles. The method described above was performed for each individual balloon flight comparison. A mean difference (with mean statistical parameters) for all flights was calculated by weighting the mean result of each individual flight equally.

Photochemically reactive gases like $NO_2$ and $N_2O_5$, and, to a lesser extent, $ClONO_2$ (mainly in the Tropics) undergo a diurnal variation with changing solar zenith angle (SZA). For these gases, a photochemical correction taking into account differences in the SZA between the





measurements of both sensors has been applied. The molecule $NO_2$ exhibits the most pronounced temporal variation. The partitioning of $NO$, $NO_2$, and $N_2O_5$ within the $NO_y$ family

depends strongly on the SZA due to the rapid photolysis of $NO_2$ and the slower photolysis of $N_2O_5$. A 1-dimensional model (Bracher et al., 2005) was constrained with $NO_y$ species measured by MIPAS-B and initialized with the output of a global 2-dimensional model (Sinnhuber et al., 2003) to calculate SZA correction factors for the MIPAS-E data.

## 220    3    Intercomparison results

In the following subsections we discuss the validation for all quantities delivered operationally by ESA's L1v8/L2v8 FM processor on the basis of collocated MIPAS-B observations. Only MIPAS satellite data that have passed the a posteriori quality check (mainly retrieval convergence and size of maximum error) were used for the intercomparison. The analysis of

all compared vertical profiles is regarded as evaluation with the highest statistical evidence. Trajectory matches are based on diabatic 2-day forward and backward trajectories with a collocation criterion of 1 h and 500 km as described in section 2.

Since the balloon flights were performed between 2003 and 2011, they cover almost the full ENVISAT operational period of 2002 to 2012, i.e., both MIPAS-E mission phases (FR and OR

modes) with distinctly different instrument settings. A compilation of all vertical profiles of temperature and 20 species retrieved from MIPAS-B spectra is given in Fig. 1. We performed the intercomparison analysis separately not only for different climatological regions but also for the periods 2002-2004 and 2005-2012. The following intercomparison is focused on these two periods when MIPAS-E was operated in the FR and OR mode, respectively. An overview

of the most important findings of this intercomparison is listed in Table 3. A comprehensive MIPAS quality readme file including MIPAS-B, ground-based and ACE-FTS (Atmospheric Chemistry Experiment – Fourier Transform Spectrometer) validation results was published by Raspollini et al. (2020).

### 3.1    Temperature

Apart from its relevance as primary atmospheric state parameter, the quality of temperature data is essential in the atmospheric emission limb sounding since temperature profiles generally are retrieved prior to the trace gas retrievals. Hence, temperature errors propagate in subsequent retrievals of trace constituents. Our study shows that above about 11 km the mean differences



between MIPAS-B and MIPAS-E are within ±2 K and within the combined systematic errors,
although the standard deviations exceed the expected precision (see Fig. 2). In the lowermost
stratosphere and around the tropopause, MIPAS-E exhibits a positive bias with respect to the
balloon instrument in the OR mode and the Tropics. Differences between both sensors are
comparable to the findings of a comprehensive temperature validation study by Ridolfi et al.
(2007) that was addressing the FR mode period only using version 4.61 and 4.62 data. However,
the large temperature differences between MIPAS-E and MIPAS in the tropical troposphere are
not seen in a comparison to groundbased data (Hubert et al., 2020). A possible reason for this
difference between both MIPAS sensors could be an inaccuracy in the altitude assignment,
which has a particularly strong effect in combination with the strong vertical temperature
gradient in the troposphere.

### 3.2   $H_2O$

In view of the ongoing debates on long-term trends of water vapour (e.g. Dessler et al., 2014;
Lossow et al., 2018; Khosrawi et al., 2018) we carefully looked at the consistency of the
validation results of the MIPAS FR phase with respect to the MIPAS OR phase. Figure 3
presents the intercomparison results. FR and OR mode comparisons show different vertical
shapes of the differences between MIPAS-E and MIPAS-B. In the lowermost stratosphere and
upper troposphere, MIPAS-E significantly overestimates $H_2O$ and exceeds the combined
systematic error bars around 15 km in the OR mode. This general behaviour remains also in the
statistical analysis of all collocations. In the middle and upper stratosphere, a positive bias of
MIPAS-E against MIPAS-B (increasing with altitude in the FR period) of 5-20 % is visible,
although the errors stay (except at 37 km) within the predicted error budget. These findings are
in line with the conclusions drawn from a comprehensive validation study of MIPAS-E (version
4.61) phase one (FR mode) observations by Wetzel et al. (2013a). The pronounced deviation
between both MIPAS sensors in the tropical troposphere may possibly be explained by an
inaccuracy in the altitude assignment in combination with the strong vertical $H_2O$ gradient in
this altitude region.

### 3.3   $O_3$

The monitoring of the expected recovery of the stratospheric ozone layer, and in particular the
Antarctic ozone hole, still remains of great scientific interest (e.g. Dhomse et al., 2019). Hence,
ozone was one of the key species during the ENVISAT mission. Comparisons based on the full





statistics over all collocations show an agreement between the satellite and the balloon data within ±10 % above 15 km for this mainly stratospheric species (see Fig. 4). For most of the stratosphere (17-37 km), the mean relative difference between the data sets is always within ±5 %. Furthermore, differences of the combined FR plus OR mode are within the combined systematic error. Degradation in the quality of the agreement is observed in the lower stratosphere and upper troposphere, with deviations up to about 20 % in both observation periods. Generally, the statistical agreement of both data sets is comparable to that reported by Cortesi et al. (2007) for the FR mode phase (v4.61/v4.62) as deduced from an extensive study using various kinds of correlative data.

### 3.4 HNO₃

$HNO_3$ is an important stratospheric nitrogen reservoir species (see e.g. Brasseur and Solomon, 2005). VMR difference profiles of this trace gas are presented in Fig. 5. MIPAS-E tends to overestimate the $HNO_3$ abundance when compared to MIPAS-B below about 27 km. This bias is most prominent in the OR mode data between 19 and 26 km around the altitude of the VMR maximum of the $HNO_3$ profile. Biases are typically in the order of 5-20 % in relative units and in line with the numbers reported by Wang et al. (2007) referring to the FR period (v4.61/v4.62). Standard deviations clearly exceed the expected precision.

### 3.5 CH₄ and N₂O

These two species are long-lived tracers of similar lifetimes and are therefore correlated to each other (see e.g. Michelsen et al., 1998). Hence, they are discussed together in this study. Figures 6 and 7 present the results for these molecules based on the statistical trajectory analysis of all collocations available. Both species show a quite similar altitude-dependent behaviour of the mean difference in absolute and relative quantities while standard deviations exceed the expected precision. MIPAS-E tends to overestimate the abundance of both species in the stratosphere below about 35 km by 5-15 % ($CH_4$) and 10-20 % ($N_2O$), respectively. A similar positive bias has already been stated in the (FR mode, v4.61) validation study by Payan et al. (2009). Somewhat larger positive deviations are visible in the Tropics around 30 km.



### 3.6  NO₂

NO$_2$ exhibits a strong diurnal variation in the stratosphere and is in photochemical equilibrium with NO and N$_2$O$_5$ (see e.g. Brasseur and Solomon, 2005). This needs to be taken into account

when comparing NO$_2$ data sets of different SZA. For our study, a photochemical correction considering differences in the SZA between the measurements of both sensors has been applied as described in more detail in section 2. Figure 8 presents the statistical trajectory match analysis. It indicates a positive bias (up to 20 %, unexplained above 31 km) of MIPAS-E NO$_2$ in the FR period that is becoming increasingly significant from lower to higher altitudes. This

is in line with the findings of the comprehensive NO$_2$ validation study (FR mode) reported by Wetzel et al. (2007) referring to v4.61 MIPAS data. In the OR period, the positive bias (above 27 km) between both sensors is smaller and amounts to about 10 %.

### 3.7  Additional v6 products: N₂O₅, ClONO₂, CFC-11, and CFC-12

Starting with processor v6, four additional target species, namely N$_2$O$_5$, ClONO$_2$, CFC-11, and

CFC-12, have been operationally processed by ESA. A first validation study of these species was carried out by Wetzel et al. (2013b).

N$_2$O$_5$ is a temporary reservoir of reactive nitrogen in the stratosphere and exhibits a prominent diurnal variation with maxima just before sunrise and minima just before sunset (see e.g. Brasseur and Solomon, 2005). The general agreement between MIPAS-E and MIPAS-B is

within ±10 % between 24 and 34 km for the mean of all collocations (see Fig. 9). Below 24 km and above 34 km, mean differences exceed at least partly the systematic errors suggesting a careful use of the MIPAS-E N$_2$O$_5$ data for scientific studies in these altitude regimes. No significant bias is visible in the OR mode, but a small negative bias is obvious in the FR period. The validation results are in line with the v6 comparison study by Wetzel et al. (2013b).

ClONO$_2$ is a major reservoir of reactive chlorine in the stratosphere and is involved in heterogeneous chemistry in the context of ozone depletion at high latitudes (e.g. Clarmann and Johansson, 2018, and references therein). It undergoes diurnal variations at higher altitudes during periods of stronger illumination, therefore it had to be photochemically corrected there. Figure 10 presents the intercomparison results for all collocations. In the altitude region where

ClONO$_2$ concentrations are most relevant, both data sets are consistent. Differences are within ±10 % between 17 and 34 km without a clear bias. Only at the upper and lower altitude edge of the comparisons, the mean differences exceed the combined systematic errors. However,



standard deviations clearly exceed the expected precision. The v8 validation results are comparable to the outcome of the study performed by Wetzel et al. (2013b) referring to v6 data.

The gases CFC-11 ($CCl_3F$) and CFC-12 ($CCl_2F_2$) are rather long-lived chlorofluorocarbons (Ko and Dak Sze, 1982). Results are shown in Figs. 11 and 12, respectively. In the case of CFC-12, mean differences remain within the combined errors and are within ±5 % (smaller than in previous versions, not shown in the plots) below 20 km. Above this altitude, a significant positive bias is visible (up to 32 km) and standard deviations exceed the expected precision.

However, this bias is less pronounced in the validation studies performed by Engel et al. (2016) and Wetzel et al. (2013b) that were based on observations in comparison to MIPAS v6 data. Deviations for the molecule CFC-11 are somewhat larger with up to ±10 % below 20 km. An increasing positive bias is obvious above this altitude level. However, CFC-11 deviations between both MIPAS instruments are smaller compared to the differences shown in the

previous validation study by Engel et al. (2016). The improvement in the quality of the CFC-11 v8 data set compared to v6 is also clearly seen in the comparisons to the MIPAS balloon observations (Wetzel et al., 2013b).

### 3.8    Additional v7 products: HCFC-22, $CCl_4$, $CF_4$, $COF_2$, and HCN

Five more species have been operationally processed by the v7 algorithm. To date, an

intercomparison study is only available for vertical VMR profiles of the molecule $CCl_4$ (Valeri et al., 2017). It should be mentioned that these species are generally more difficult to retrieve than the gases described before. This holds also for the MIPAS-B retrieval, although these gases can be measured with higher accuracy (mainly due to lower spectral noise) compared to MIPAS-E. Hence, some unexplained features (exceeding combined systematic errors) in the

VMR difference profiles are expected to occur more frequently when comparing these molecules.

HCFC-22 ($CHClF_2$) is a longer-lived hydrochlorofluorocarbon. Since HCFC-22 is often used as an alternative to the highly ozone-depleting CFC-11 and CFC-12, its tropospheric concentration is further increasing (e.g. Chirkov et al., 2016). Comparison results are depicted

in Fig. 13. In the FR mode period, differences between both instruments remain within ±10 % up to 26 km turning into a significant positive bias above this altitude. In the OR observation period, deviations stay within 10 % for altitudes up to 28 km while a significant negative bias





is visible in the MIPAS-E data above this altitude level. Standard deviations exceed the expected precision at higher altitudes (mainly OR phase).

The tropospheric mixing ratio of the longer-lived source gas $CCl_4$ is clearly decreasing since the beginning of the 1990s (Prinn et al., 2000). However, estimated sources and sinks of this molecule are inconsistent with observations of its abundance (Carpenter et al., 2014). A significant negative bias shows up in the MIPAS-E $CCl_4$ data (full period) above 22 km (see Fig. 14), which is at the brink of the combined systematic error limits. A significant positive

bias is visible below 21 km during the OR phase. However, differences stay within ±20 % up to about 22 km in both observation periods, which is in line with the deviations reported by Valeri et al. (2017) referring to v7 data.

The fluorocarbon $CF_4$ has an extremely long atmospheric lifetime of more than 50000 years and its atmospheric concentration is linearly increasing (Carpenter et al., 2014). Comparison

results are shown in Fig. 15. A general agreement between both instruments can be stated between 11 and 37 km (within ±10 % in both observation periods). In the FR phase, a significant positive bias above 10 km is visible. In contrast, no clear bias is obvious in the OR period where differences stay within ±10 % at all altitudes. However, standard deviations exceed the expected precision in the OR phase.

The molecule $COF_2$ is a stratospheric reservoir species for fluorine (Harrison et al., 2014). The general profile shape (as measured by MIPAS-B) is reproduced by MIPAS-E (see Fig. 16). Stratospheric VMR differences stay within ±10 % in the FR period and ±20 % in the OR period. No unexplained biases (in terms of combined systematic error bars) are evident.

HCN is mainly produced by biomass burning and hence considered as an almost unambiguous

tracer for biomass burning events (e.g. Li et al., 2003). Differences are within ±20 % below 34 km (see Fig. 17). A significant positive bias (more than 20 %) is evident in the MIPAS-E profiles observed in the FR mode period exceeding the combined systematic error limits above 20 km. This pronounced bias is visible in each comparison of the three MIPAS-B flights in the FR phase. No clear bias can be seen in the OR period. The standard deviation between about

20 km and 30 km exceeds the estimated precision in the OR phase.

### 3.9 Additional v8 products: $C_2H_2$, $C_2H_6$, $COCl_2$, OCS, and $CH_3Cl$

Some more target molecules have been operationally processed by the v8 algorithm. To date, an intercomparison study is only available for vertical VMR profiles of the species $COCl_2$





(Pettinari et al., 2021). Similar to the additional v7 gases, the emissions of spectral lines of the

v8 molecules are also weak compared to the spectral signatures of the standard gases (before

v7). Hence, retrievals of these additional species are challenging.

$C_2H_2$ is mainly produced by biomass burning and, to a lesser extent, by biofuel burning (e.g.

Singh et al., 1996; Parker et al., 2011; Wiegele et al., 2012). Differences are within ±50 % up

to 24 km (see Fig. 18). A significant negative bias (within -50 % difference limit) is evident in

the FR mode (except for 15-16 km). A significant negative bias below 20 km and above 23 km

can be seen in the OR mode (exceeding combined systematic errors and the -50 % difference

limit). Lower stratospheric altitude regions in MIPAS-E retrievals sometimes show negative

VMRs (in Arctic winter). Hence, this species should be carefully used in scientific studies.

$C_2H_6$ is produced by biomass burning, natural gas losses and fossil fuel production (e.g.

Rudolph, 1995; Xiao et al., 2008; Glatthor et al., 2009). Differences are within ±25 % up to

19 km (see Fig. 19). While a significant negative bias is obvious in the FR period (exceeding

the -50 % limit above 13 km), no bias is seen in the MIPAS-E data below 20 km in the OR

mode, where differences are within a ±20 % range. Lower stratospheric altitude regions in

MIPAS-E retrievals sometimes show negative VMRs (in the Arctic). Consequently, $C_2H_6$

profiles should be carefully used in scientific studies.

$COCl_2$ is produced by chemical industries and OH-initiated oxidation of chlorinated

hydrocarbons in the troposphere (Kindler et al., 1995; Fu et al., 2007; Valeri et al., 2016). Figure

20 shows that differences are within ±20 % up to 27 km in both observation periods, such that

the general profile shapes (as measured by MIPAS-B) are reproduced by the satellite

instrument. A negative bias is evident in the FR and OR period (except for 22-27 km),

unexplained at high altitudes. Deviations in the Tropics are quite large. The deviations are in

line with the findings of Pettinari et al. (2021) who compared v8 data not only to MIPAS-B but

also to observations from ACE-FTS. Pettinari et al. (2021) found that some of the differences

between MIPAS-E and MIPAS-B can be attributed to the different spectroscopic data used

(Toon et al. (2001) for MIPAS-B and Tchana et al. (2015) in the case of MIPAS-E).

OCS is the most prevalent sulphur-containing species which is transported into the stratosphere

where it acts as prerequisite of the stratospheric aerosol layer (Crutzen, 1976; Kremser et al.,

2016; Glatthor et al., 2017). Differences are within ±20 % up to 24 km in the FR period and

within ±25 % up to 25 km in the OR period (see Fig. 21). A significant positive bias is visible

below 22 km and a negative bias above this altitude in the OR period exceeding the ±50 % limit





and the combined systematic errors above 24 km. The agreement of the VMR profiles of both sensors is better in the FR period. Here, a significant (positive) bias is only visible between 14 and 18 km. Deviations in the Tropics are quite large.

$CH_3Cl$ is the most abundant halocarbon in the atmosphere and originates from natural and anthropogenic sources (see e.g. Yokouchi et al., 2000). Fig. 22 shows that differences stay within ±20 % between 13 and 22 km (full observation period). However, the comparison reveals a positive bias above 16 km and a negative bias below this altitude in the FR period. A negative bias within -35 % between 19 and 26 km, increasing with altitude, and exceeding the combined systematic errors above 26 km is also visible in the OR period. Large deviations between both instruments occur at midlatitudes and in the Tropics.

## 4 Conclusions

Vertical profiles of MIPAS balloon flights between 2002 and 2011 covering virtually the whole lifetime of MIPAS on ENVISAT have been used for an intercomparison study of all operational parameters delivered by ESA (except HDO), namely temperature and 20 species as listed in Table 1. The main findings of this intercomparison study are summarized in Table 3. The difference between retrieved temperature profiles of both MIPAS instruments generally stays within ±2 K in the stratosphere. The MIPAS satellite observations of a large number of gases like $H_2O$, $O_3$, $HNO_3$, $CH_4$, $N_2O$, $NO_2$, $N_2O_5$, $ClONO_2$, CFC-11, CFC-12, HCFC-22, $CCl_4$, $CF_4$, $COF_2$, and HCN show an overall good agreement of 5 % to 20 % with the MIPAS balloon measurements in the lower stratosphere.

The intercomparison of the new MIPAS v8 products $C_2H_2$, $C_2H_6$, $COCl_2$, OCS, and $CH_3Cl$ exhibits a somewhat reduced agreement with the MIPAS-B observations compared to the above-mentioned species. However, $COCl_2$, OCS, and $CH_3Cl$ achieve a 20-percent agreement at least in the extratropical upper troposphere and lower stratosphere.

Overall it can be stated that the v8 operational MIPAS data can be recommended for scientific use. However, data users are strongly advised to consider the findings presented in this study in the respective sections and in Table 3 when using the MIPAS data. A comprehensive MIPAS quality readme file including MIPAS-B, ground-based and ACE-FTS validation results was published by Raspollini et al. (2020) and is recommended for data users who want to get more detailed information on the quality of MIPAS data.



*Data availability.* MIPAS operational satellite data are available at the European Space Agency mission web page (https://doi.org/10.5270/EN1-c8hgqx4, last access: 12 July 2022). MIPAS

balloon data are available upon request (https://www.imk-asf.kit.edu/ffb.php, last access: 12 July 2022).

*Author contributions.* GW wrote the paper and performed the bulk of the MIPAS balloon data analysis, with input from all co-authors. AK performed the MIPAS-B Level 1 data processing.

MH provided the retrieval software. FFV and GM operated the balloon instrument during all campaigns. HO directed the research and flight planning. MS used a 1-dimensional model for photochemical corrections. JA performed trajectory match calculations. PR had a leading role in evaluating and improving the MIPAS ESA data. AD was the head of the MIPAS Quality Working Group and coordinated the validation activities. All authors commented on and

improved the manuscript.

*Competing interests.* The authors declare that they have no conflict of interest.

*Special issue statement.* This article is part of the special issue "MIPAS ESA Level 2 version 8

products: algorithms, product features and validation". It is not associated with a conference.

*Acknowledgements.* Financial support by the DLR (Project 50EE0020) and ESA for the MIPAS balloon flights is gratefully acknowledged. We thank the Centre National d'Etudes Spatiales (CNES) balloon launching team and the Swedish Space Corporation (SSC) Esrange team for

excellent balloon operations. An acknowledgement goes to the work performed by the Quality Working Group established by ESA for verification and monitoring of MIPAS products. A corresponding report on the validation activities was published by ESA (Wetzel et al., 2020). We acknowledge support by Deutsche Forschungsgemeinschaft and the Open Access Publishing Fund of Karlsruhe Institute of Technology.






**Table 1.** Overview of MIPAS-B spectral windows used for the analysis of atmospheric target parameters together with typical precision errors and total errors.

| Target parameter | Spectral range (cm$^{-1}$) | Precision error | Total error |
|---|---|---|---|
| **Temperature** | 801.1 – 813.2<br>941.3 – 956.7 | 0.2 – 0.3 K | 0.5 – 1.0 K |
| **H$_2$O** | 808.0 – 825.3<br>1210.2 – 1244.5<br>1585.0 – 1615.0 | 1 – 2 % | 8 – 11 % |
| **O$_3$** | 763.5 – 824.4<br>964.9 – 969.0<br>1140.1 – 1195.6 | 0.1 – 1 % | 8 – 10 % |
| **HNO$_3$** | 864.0 – 874.0 | 0.2 – 2 % | 8 – 9 % |
| **CH$_4$ & N$_2$O** | 1161.9 – 1229.8 | 1 – 3 % | 6 – 10 % |
| **NO$_2$** | 1585.0 – 1615.0 | 1 – 3 % | 10 – 12 % |
| **N$_2$O$_5$** | 1220.0 – 1270.0 | 0.4 – 2 % | 5 – 7 % |
| **ClONO$_2$** | 779.7 – 780.7 | 2 – 3 % | 5 – 6 % |
| **CFC-11** | 840.0 – 860.0 | 2 – 3 % | 5 – 6 % |
| **CFC-12** | 918.0 – 924.0 | 2 – 3 % | 5 – 6 % |
| **HCFC-22** | 828.0 – 830.0 | 3 – 6 % | 9 – 12 % |
| **CCl$_4$** | 786.0 – 806.0 | 5 – 10 % | 11 – 15 % |
| **CF$_4$** | 1274.3 – 1288.0 | 2 – 6 % | 6 – 11 % |
| **COF$_2$** | 750.0 – 776.0 | 1 – 3 % | 10 – 12 % |
| **HCN** | 750.0 – 776.0 | 4 – 8 % | 9 – 12 % |
| **C$_2$H$_2$** | 750.2 – 790.1 | 5 – 10 % | 7 – 12 % |
| **C$_2$H$_6$** | 811.5 – 835.8 | 8 – 12 % | 12 – 15 % |
| **COCl$_2$** | 838.3 – 860.0 | 2 – 5 % | 20 – 22 % |
| **OCS** | 842.4 – 876.0 | 15 – 20 % | 18 – 25 % |
| **CH$_3$Cl** | 742.5 – 755.0 | 5 – 15 % | 12 – 20 % |






**Table 2.** Overview of MIPAS balloon flights used for intercomparison with MIPAS-E. Distances and times between closest trace gas profile pairs observed by MIPAS-E and the validation instrument refer to an altitude of 20 km (Kiruna) and 30 km (Aire sur l'Adour and Teresina). In addition, 2-day forward/backward trajectories were calculated for each balloon flight to search for further matches with the satellite sensor.

| Location | Date | Distance (km) | Time difference (min) |
|---|---|---|---|
| **Kiruna, 68 °N** | 20 Mar 2003 | 16 / 546 | 14 / 15 |
| | 03 Jul 2003 | Trajectories only | |
| | 11 Mar 2009 | 187 / 248 | 5 / 6 |
| | 24 Jan 2010 | 109 / 302 | 5 / 6 |
| | 31 Mar 2011 | Trajectories only | |
| **Aire sur l'Adour, 44 °N** | 24 Sep 2002 | 21 / 588 / 410 / 146 | 12 /13 / 15 / 16 |
| **Teresina, 5 °S** | 14 Jun 2005 | 109 / 497 / 184 / 338 | 228 / 229 / 268 / 269 |
| | 06 Jun 2008 | 224 / 284 / 600 / 194 | 157 / 158 / 169 / 170 |






**Table 3.** Summary of MIPAS-E validation results (trajectory comparison to eight MIPAS-B flights). Mentioned atmospheric parameter differences refer to MIPAS-E minus the balloon instrument.

| Parameter | Comments (L1v8/L2v8 FM) |
|---|---|
| Temp. | Differences within ±2 K between 12 and 39 km. |
| $H_2O$ | Positive bias (5-20 %) between 11 and 39 km within combined systematic errors (except OR mode around 15 km). |
| $O_3$ | Differences within ±10 % for all altitudes above 15 km. |
| $HNO_3$ | Significant positive bias (5-20 %) below 27 km (most pronounced between 19 and 26 km in the OR mode). |
| $CH_4$ & $N_2O$ | Positive bias for $CH_4$ (5-15 %) and $N_2O$ (10-20 %) below 35 km, especially pronounced for $N_2O$ in the lowermost stratosphere around 15 km. Somewhat larger positive deviations also in the Tropics around 30 km. |
| $NO_2$ | Positive bias of up to 20 % in FR mode (unexplained above 31 km), smaller positive bias (~10 %) in OR mode (above 27 km). |
| $N_2O_5$ | Differences within ±10 % between 24 and 34 km (no significant bias in OR mode, small negative bias in FR period). |
| $ClONO_2$ | Differences within ±10 % between 17 and 34 km (no significant bias). |
| CFC-11 | Differences within 10 % below 20 km. Positive bias (increasing with altitude) above this altitude level. |
| CFC-12 | Differences within ±5 % for altitudes below 20 km. Significant positive bias above this altitude level up to 32 km. |
| HCFC-22 | Differences within ±10 % up to 26 km (FR mode) and 28 km (OR mode). Positive differences up to 20 % above 26 km (FR mode) and significant negative bias above 28 km (OR mode). |
| $CCl_4$ | Differences within ±20 % up to about 22 km in both observation periods. Increasing negative bias above 22 km (full period). |
| $CF_4$ | Differences within ±10 % between 11 and 37 km (both periods). Significant positive bias above 10 km in FR period. No clear bias in OR period. |
| $COF_2$ | Differences within ±10 % for FR period and within ±20 % in OR period in the stratosphere; no unexplained biases. |
| HCN | Differences within ±20 % below 34 km. Stratospheric positive bias in FR mode, exceeding combined systematic errors above 20 km (difference > 20 %). No clear bias in OR period. |
| $C_2H_2$ | Differences within ±50 % up to 24 km. Negative bias (within 50 %) in FR mode (except 15-16 km), significant negative bias below 20 km and above 23 km in OR mode (exceeding combined systematic errors and the -50 % difference limit). Lower stratospheric altitude regions in MIPAS-E retrievals sometimes show negative VMRs (in Arctic winter). |
| $C_2H_6$ | Differences within ±25 % up to 19 km. Significant negative bias in FR mode (exceeding -50 % limit above 13 km), no bias in OR mode below 20 km (differences within ±20 %). Lower stratospheric altitude regions in MIPAS-E retrievals sometimes show negative VMRs (in the Arctic). |
| $COCl_2$ | Differences within ±20 % up to 27 km in both periods. Negative bias in FR and OR period (except 22-27 km), unexplained at high altitudes; quite large deviations in the Tropics. Parts of differences can be attributed to new spectroscopic data (MIPAS-E retrieval). |
| OCS | Differences within ±20 % up to 24 km in FR period. Significant positive bias between 14 and 18 km; difference within ~25 % up to 25 km (OR mode). Significant positive bias < 22 km and negative bias > 22 km (OR mode) exceeding ±50 % limit and combined systematic errors above 24 km; quite large deviations in the Tropics. |
| $CH_3Cl$ | Differences within ±20 % between 13 and 22 km. Positive bias above 16 km (negative bias below) in FR period. Negative bias within -35 % between 19 and 26 km, increasing with altitude, and exceeding the combined systematic errors above 26 km (OR period). Large deviations at midlatitudes and in the Tropics. |



**Figure 1.** Retrieved vertical profiles of temperature **(a)** and species **(b-u)** of Arctic winter (blue), Arctic summer (cyan), midlatitude (green) and tropical (red) MIPAS-B flights as listed in Table 2.



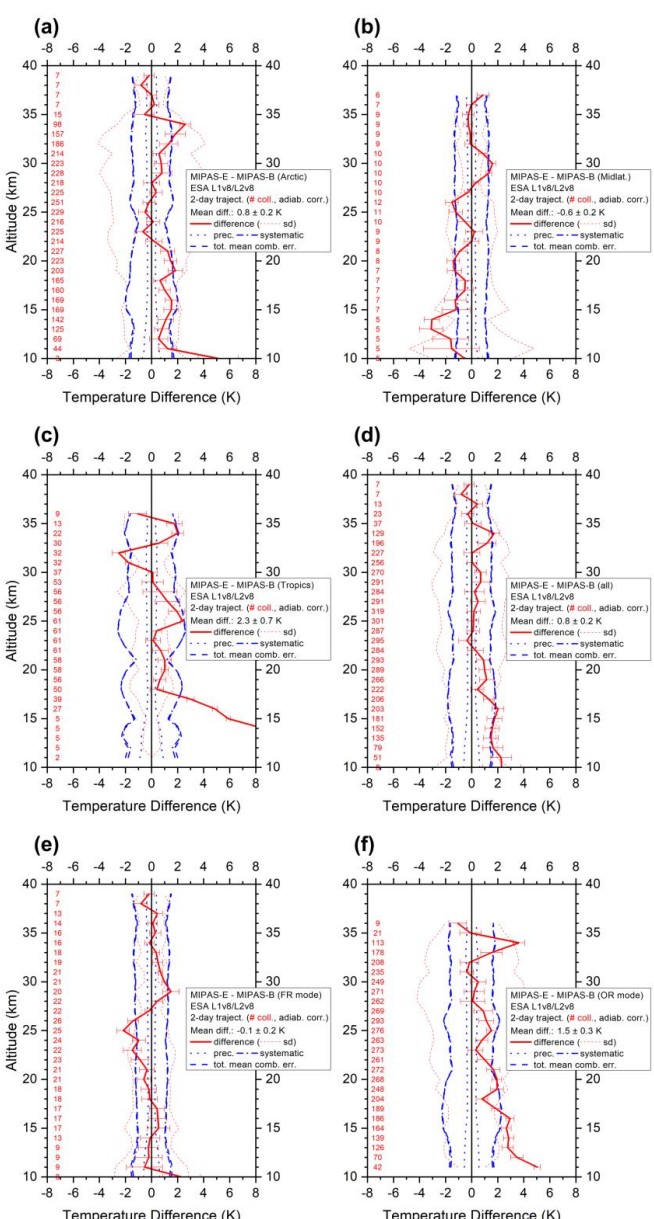

**Figure 2.** Mean temperature difference (red solid line) of all trajectory match collocations (red numbers) between MIPAS-E and MIPAS-B including standard deviation (red dotted lines) and standard error of the mean (plotted as error bars). Precision (blue dotted lines), systematic (blue dash-dotted lines), and total (blue dashed lines) mean combined errors are shown, too. Arctic **(a)**, midlatitude **(b)**, Tropics **(c)**, all FR plus OR **(d)**, FR mode **(e)**, and OR mode **(f)** collocations. For details, see text.

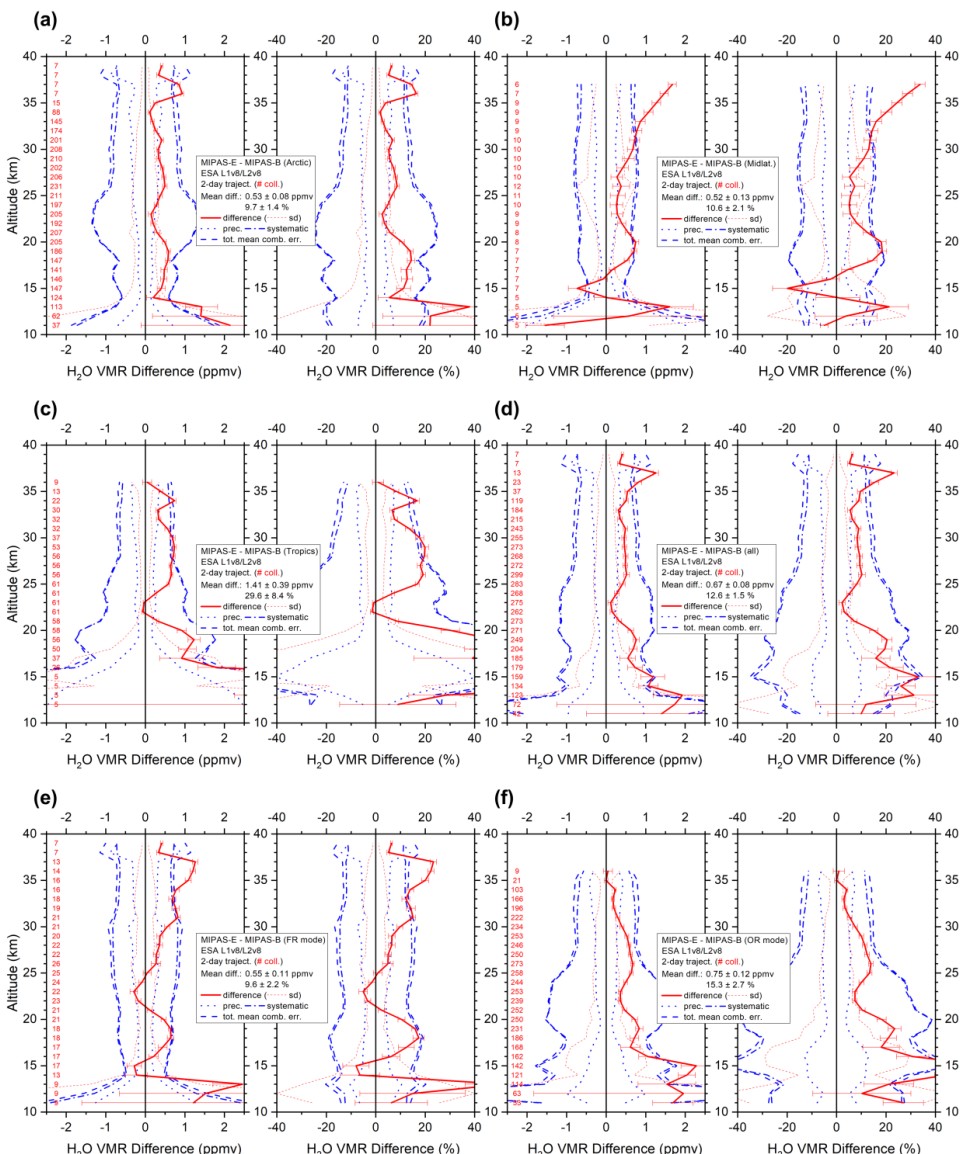

**Figure 3.** Mean absolute and relative H$_2$O VMR difference of all trajectory match collocations (red numbers) between MIPAS-E and MIPAS-B (red solid line) including standard deviation (red dotted lines) and standard error of the mean (plotted as error bars). Precision (blue dotted lines), systematic (blue dash-dotted lines), and total (blue dashed lines) mean combined errors are shown, too. Arctic **(a)**, midlatitude **(b)**, Tropics **(c)**, all FR plus OR **(d)**, FR mode **(e)**, and OR mode **(f)** collocations. For details, see text.

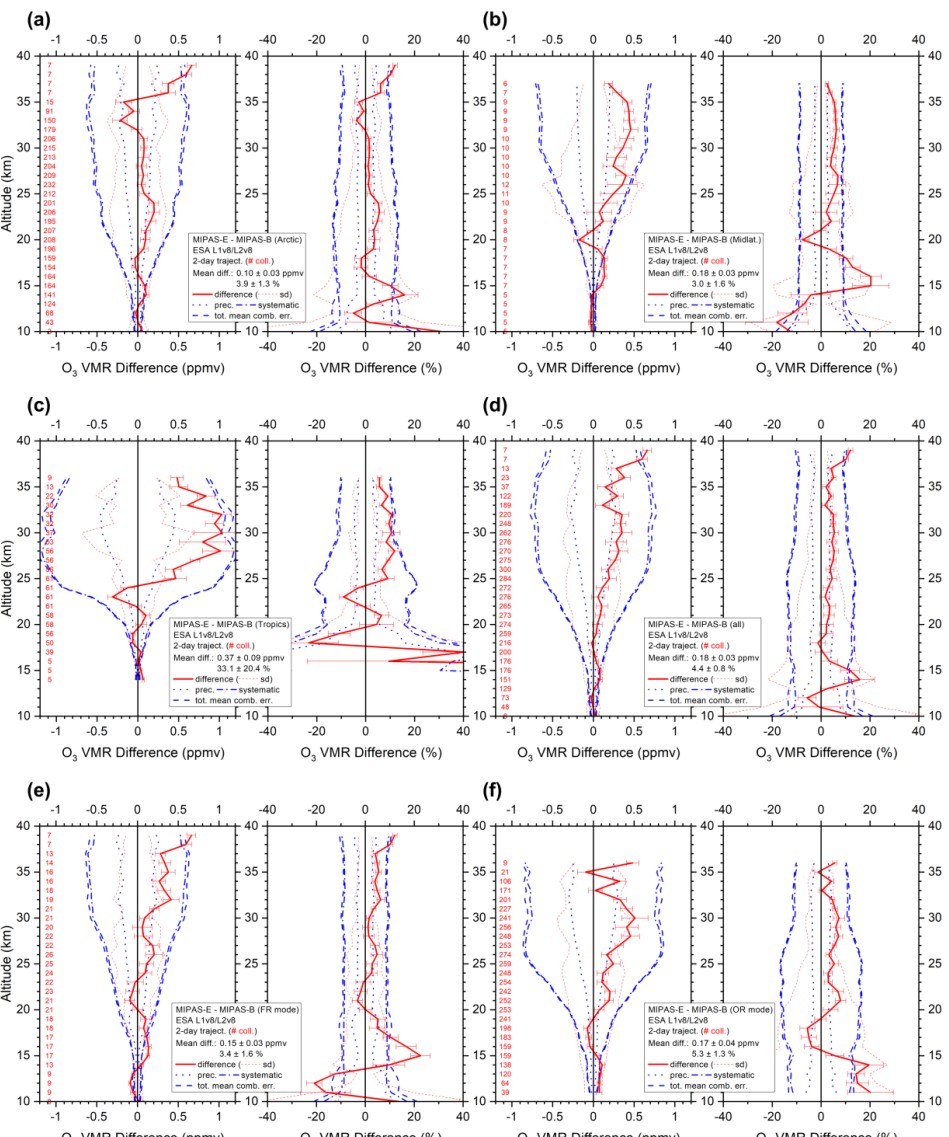

**Figure 4.** Same as Fig. 3 but for $O_3$.



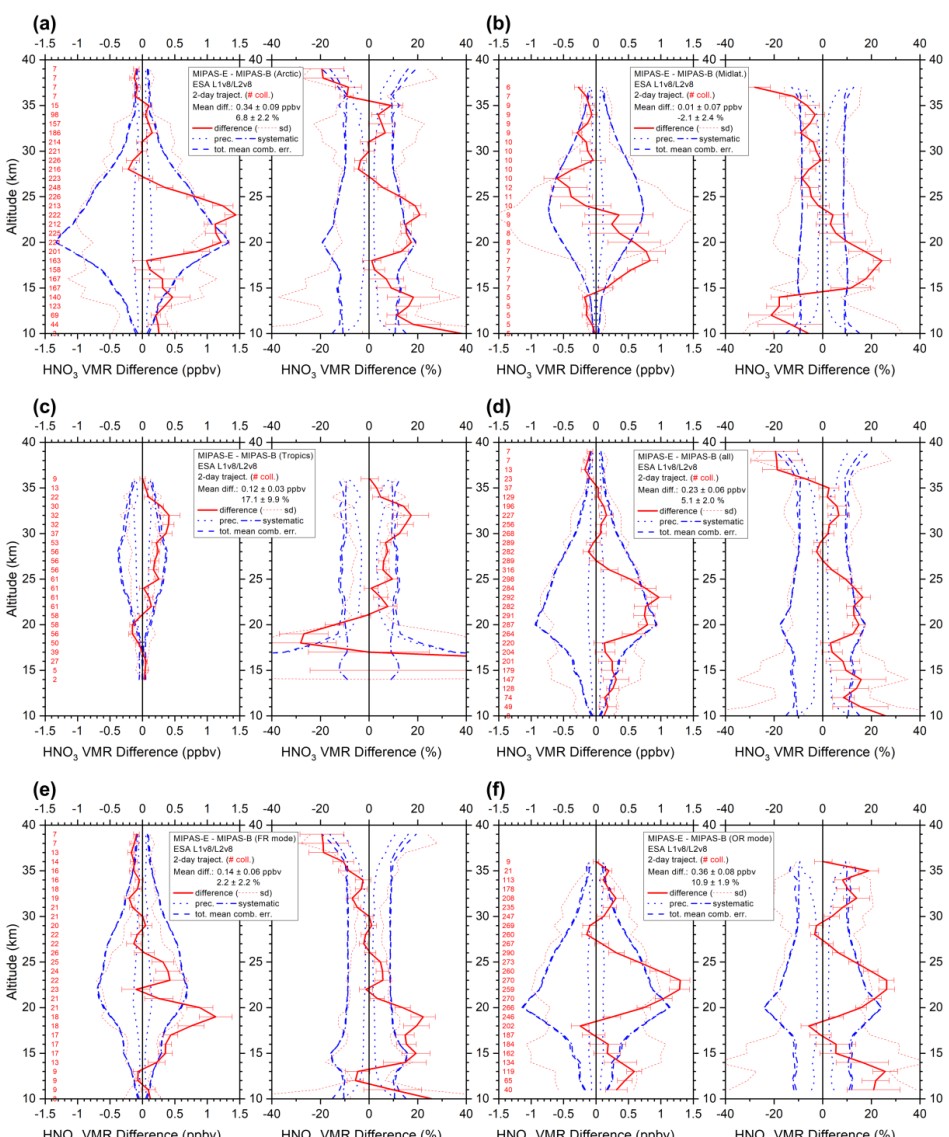

**Figure 5.** Same as Fig. 3 but for HNO$_3$.




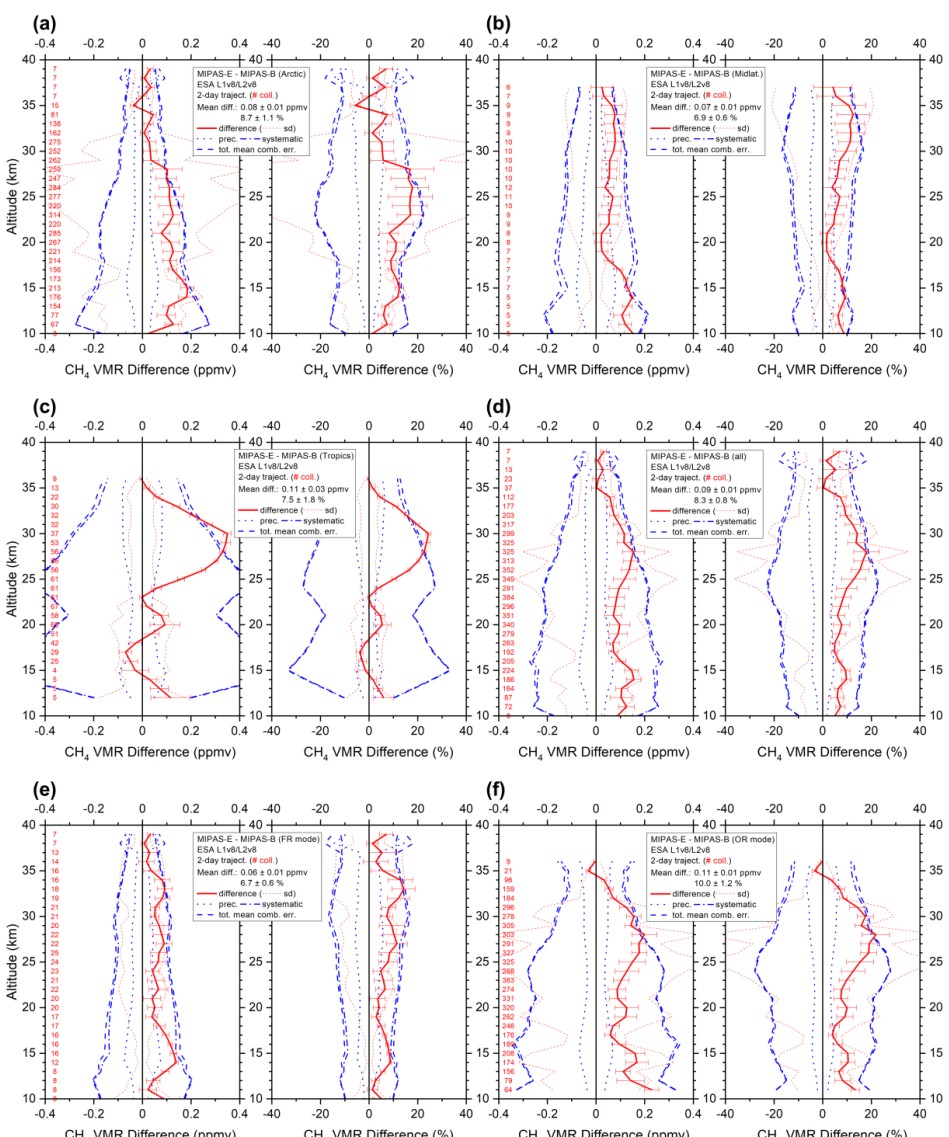

**Figure 6.** Same as Fig. 3 but for CH$_4$.

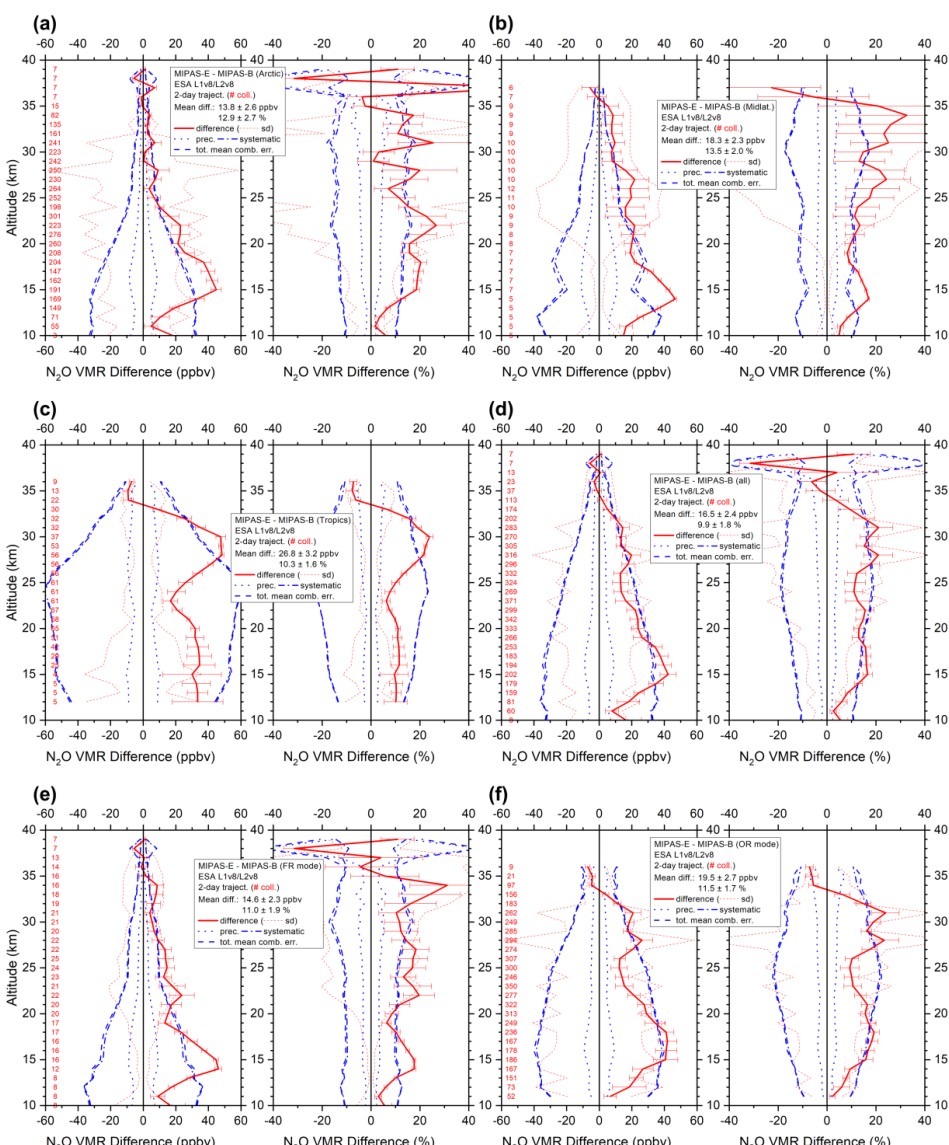

**Figure 7.** Same as Fig. 3 but for $N_2O$.



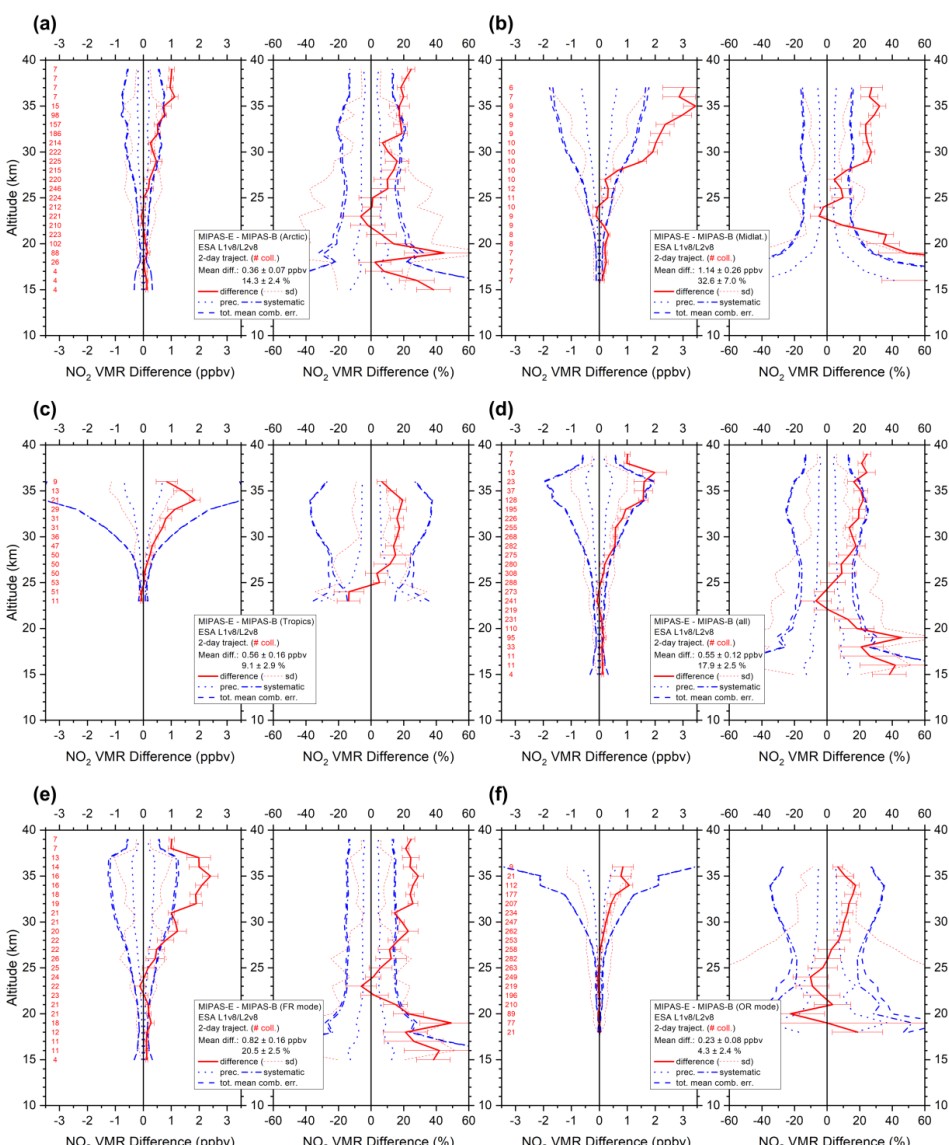

**Figure 8.** Same as Fig. 3 but for NO$_2$.



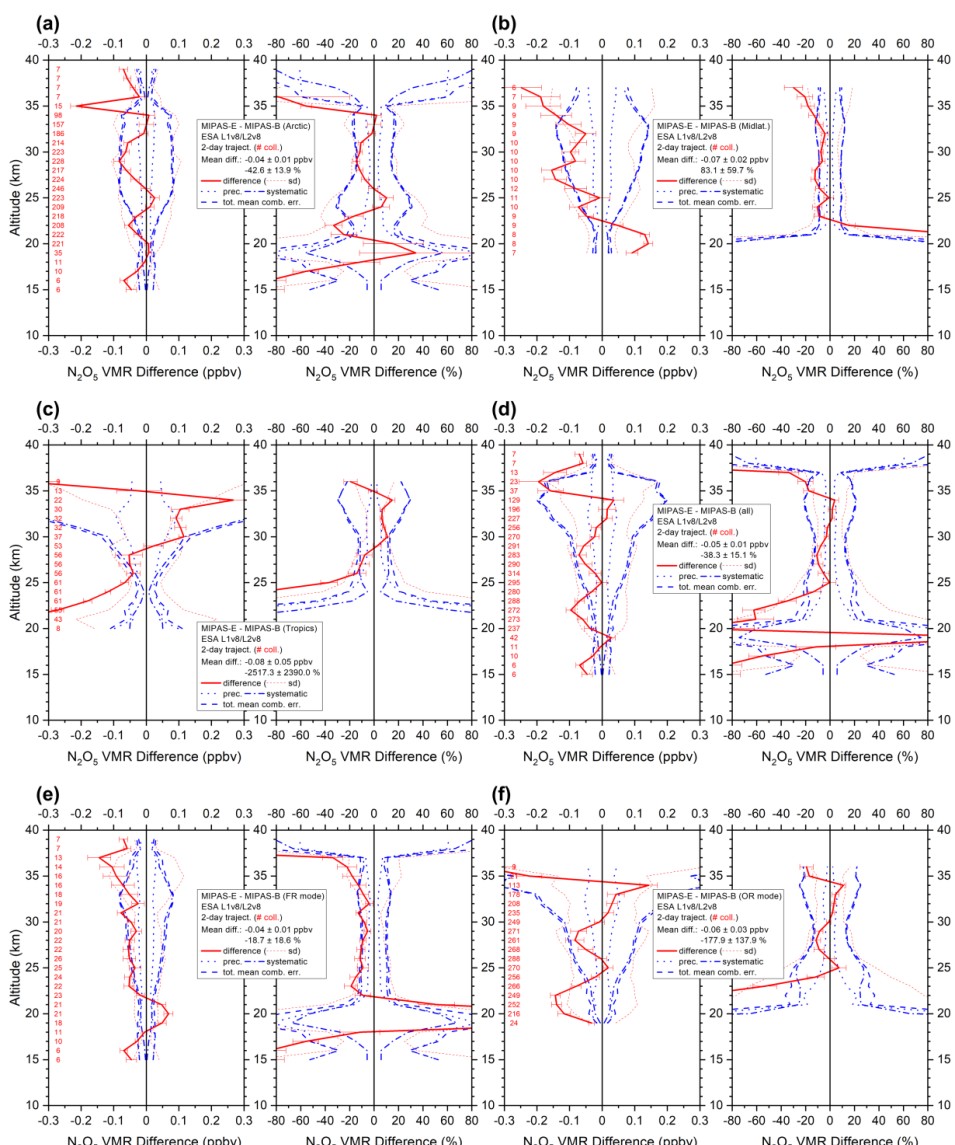

**Figure 9.** Same as Fig. 3 but for $N_2O_5$.



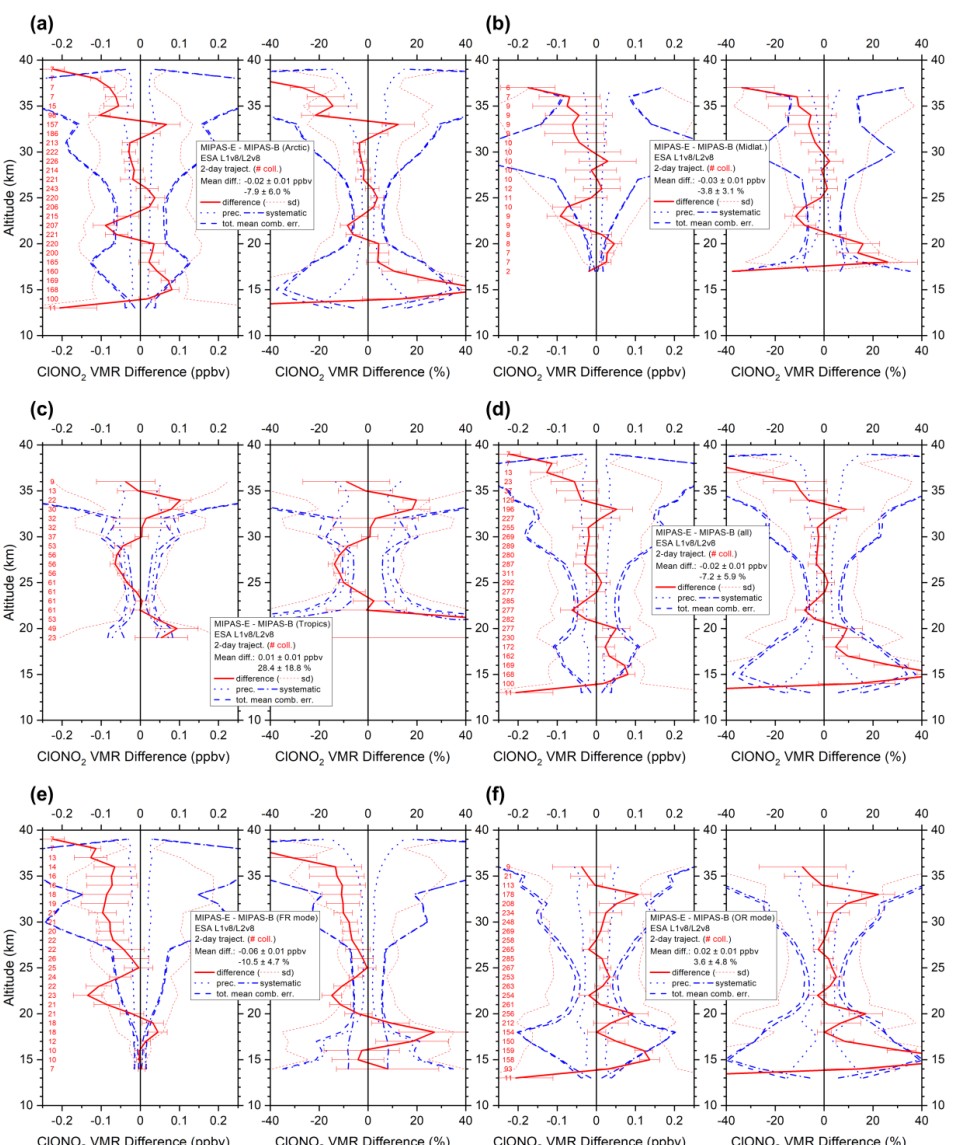

**Figure 10.** Same as Fig. 3 but for ClONO$_2$.


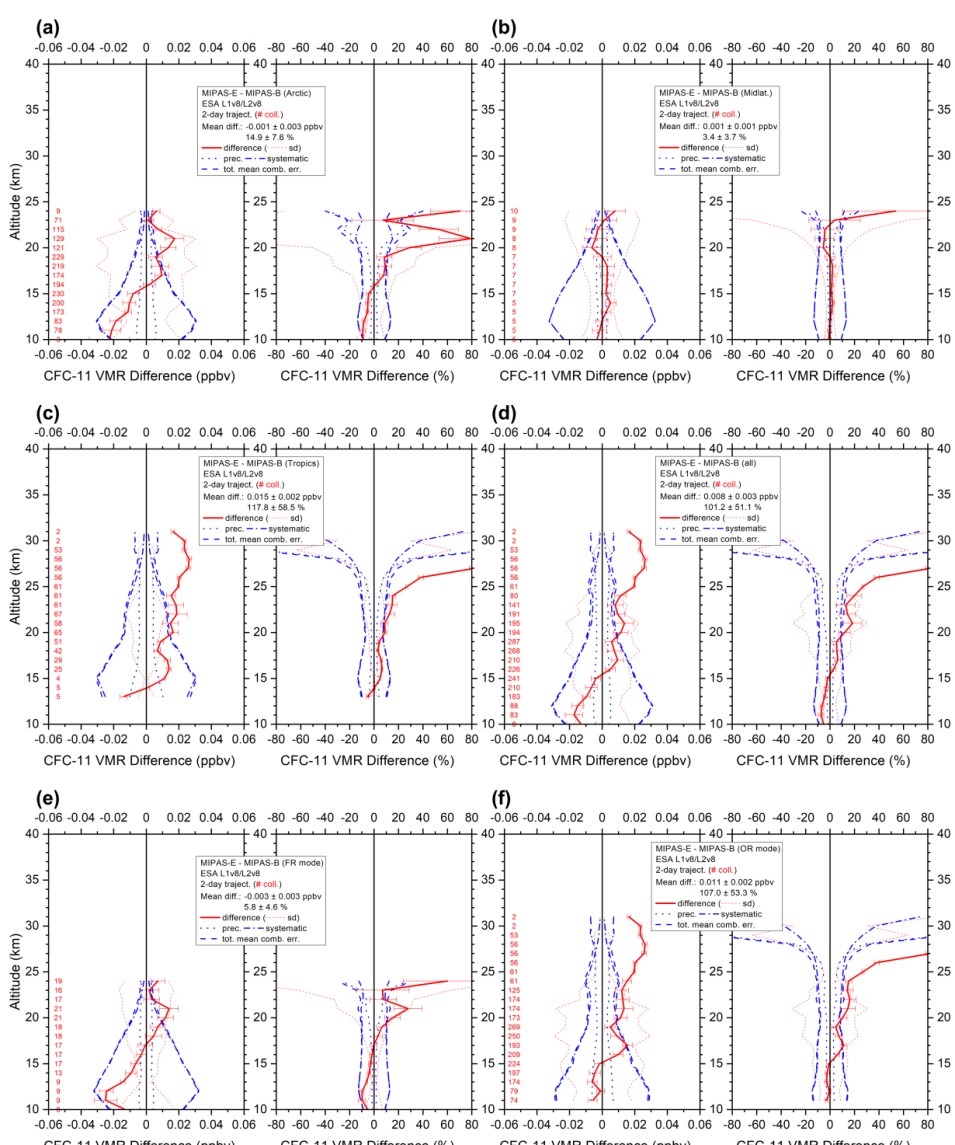

**Figure 11.** Same as Fig. 3 but for CFC-11.



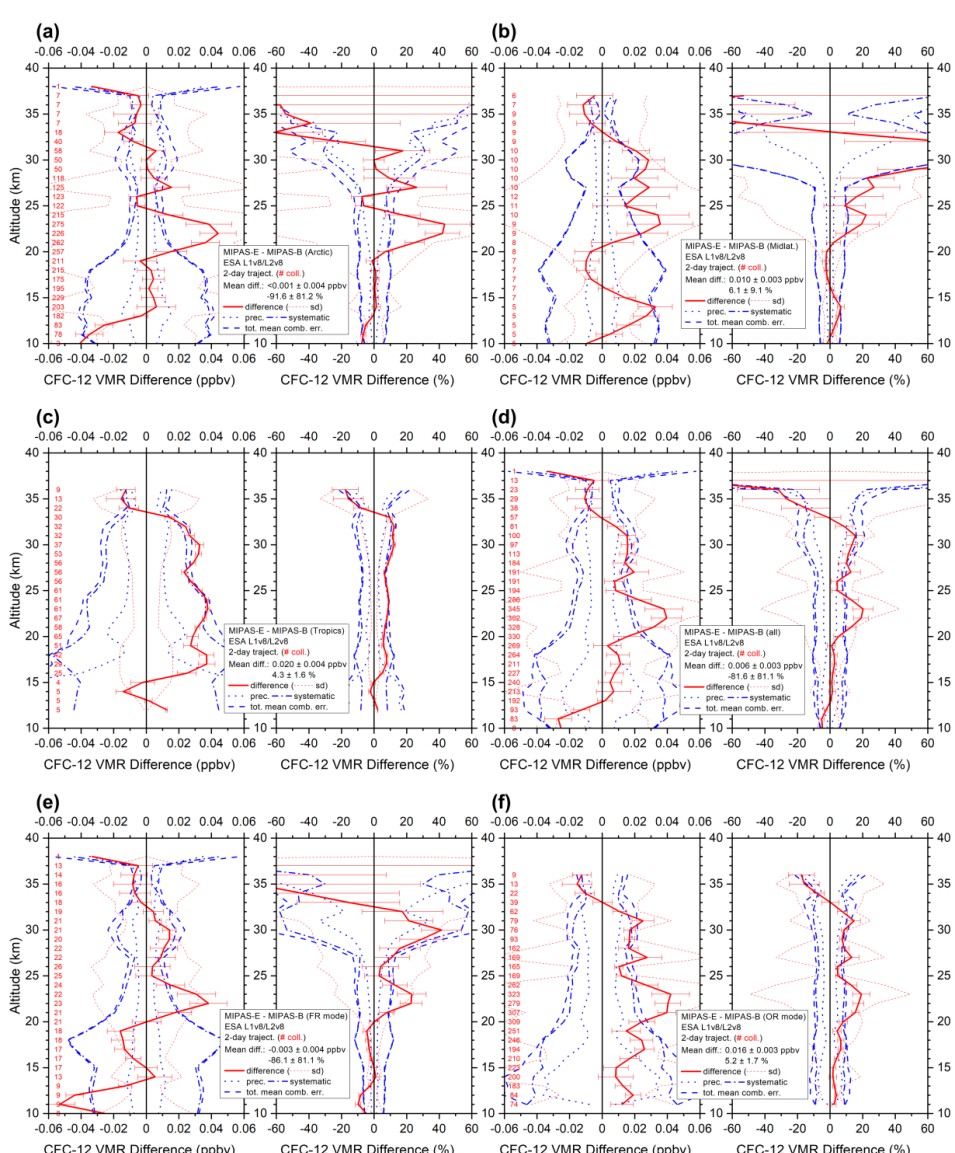

**Figure 12.** Same as Fig. 3 but for CFC-12.

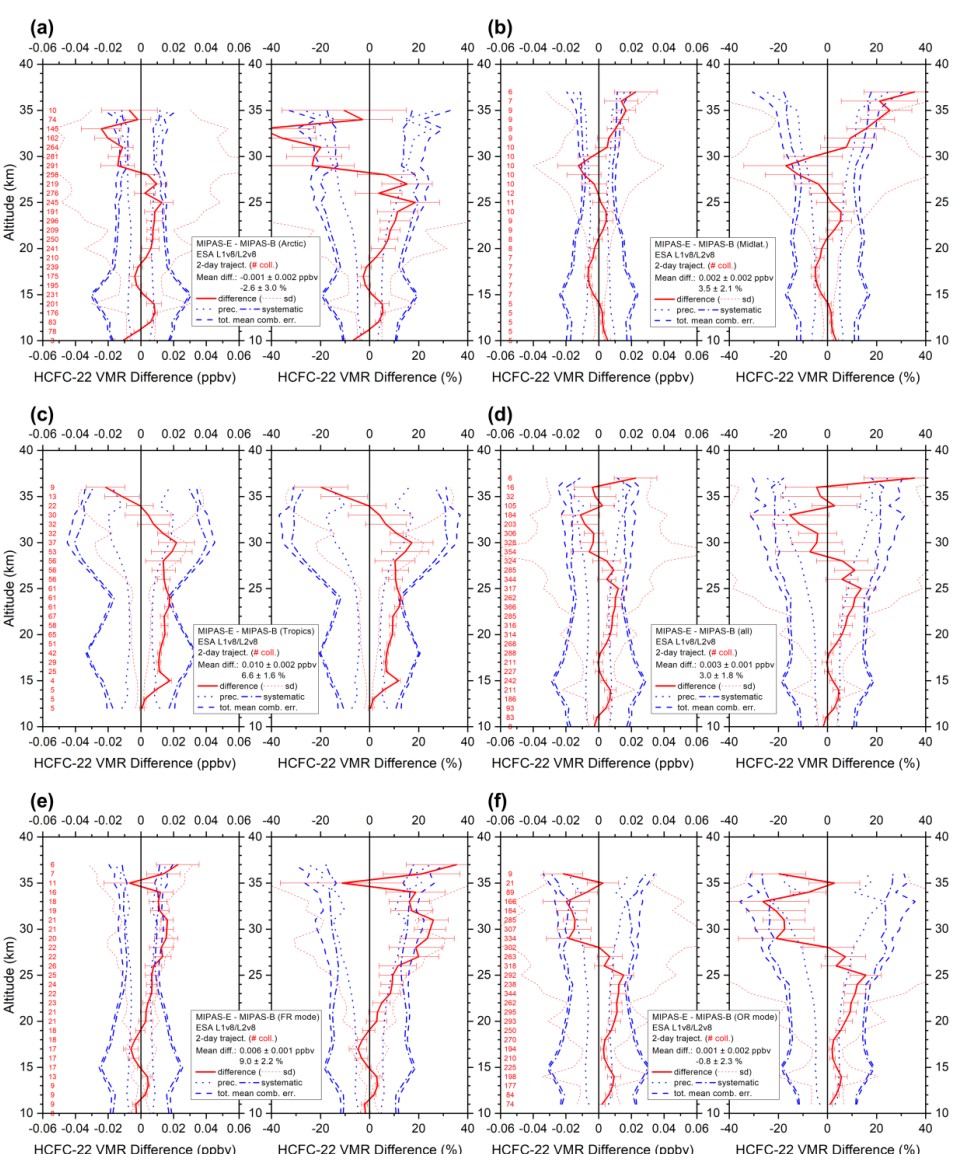

**Figure 13.** Same as Fig. 3 but for HCFC-22.



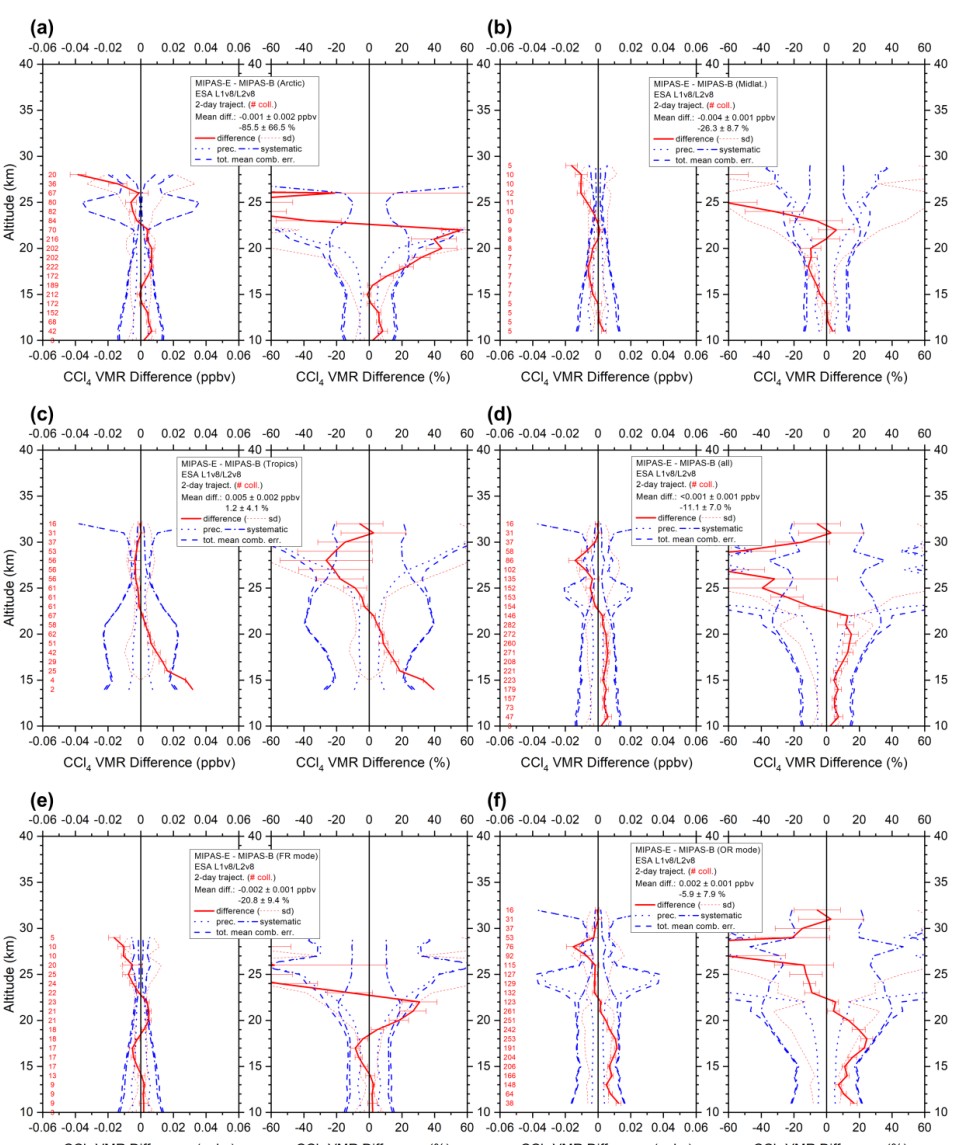

**Figure 14.** Same as Fig. 3 but for CCl₄.



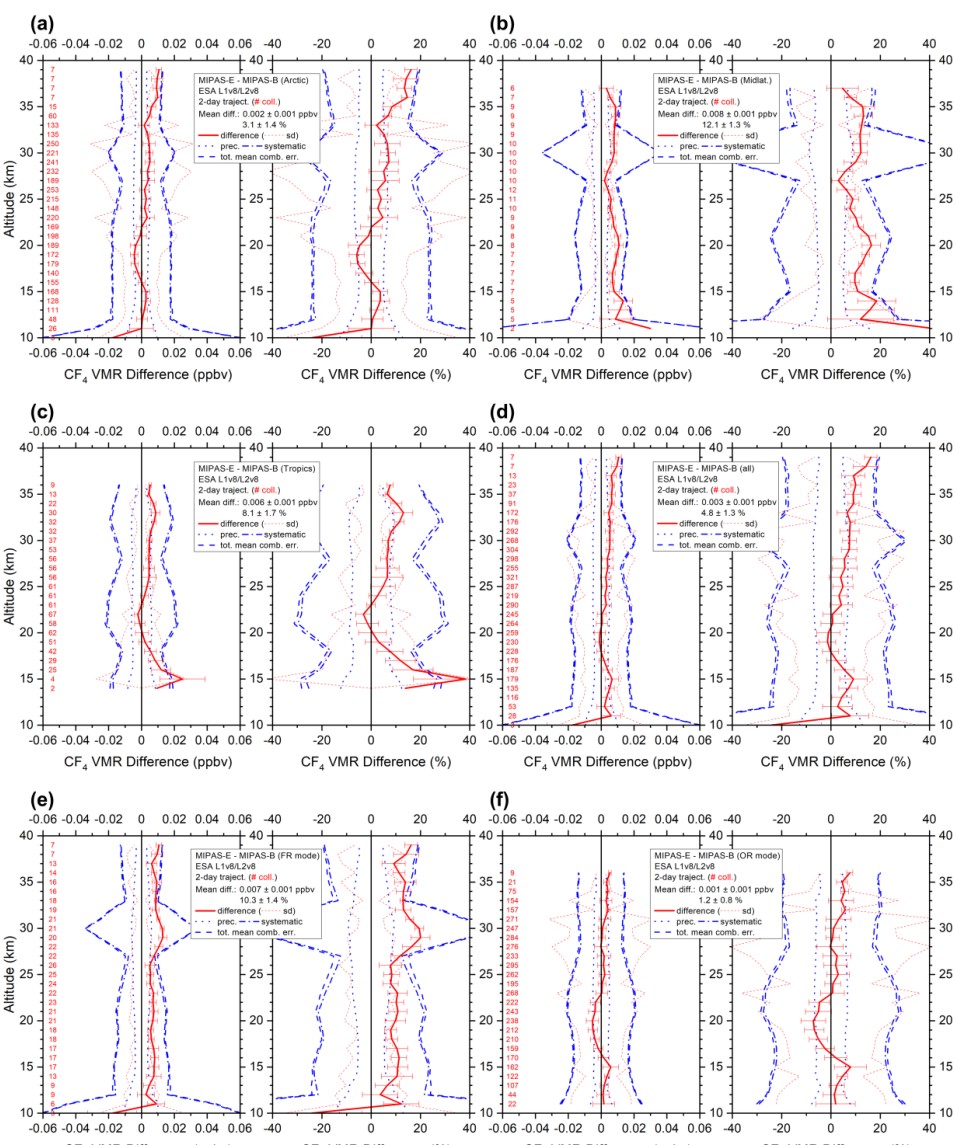

**Figure 15.** Same as Fig. 3 but for $CF_4$.




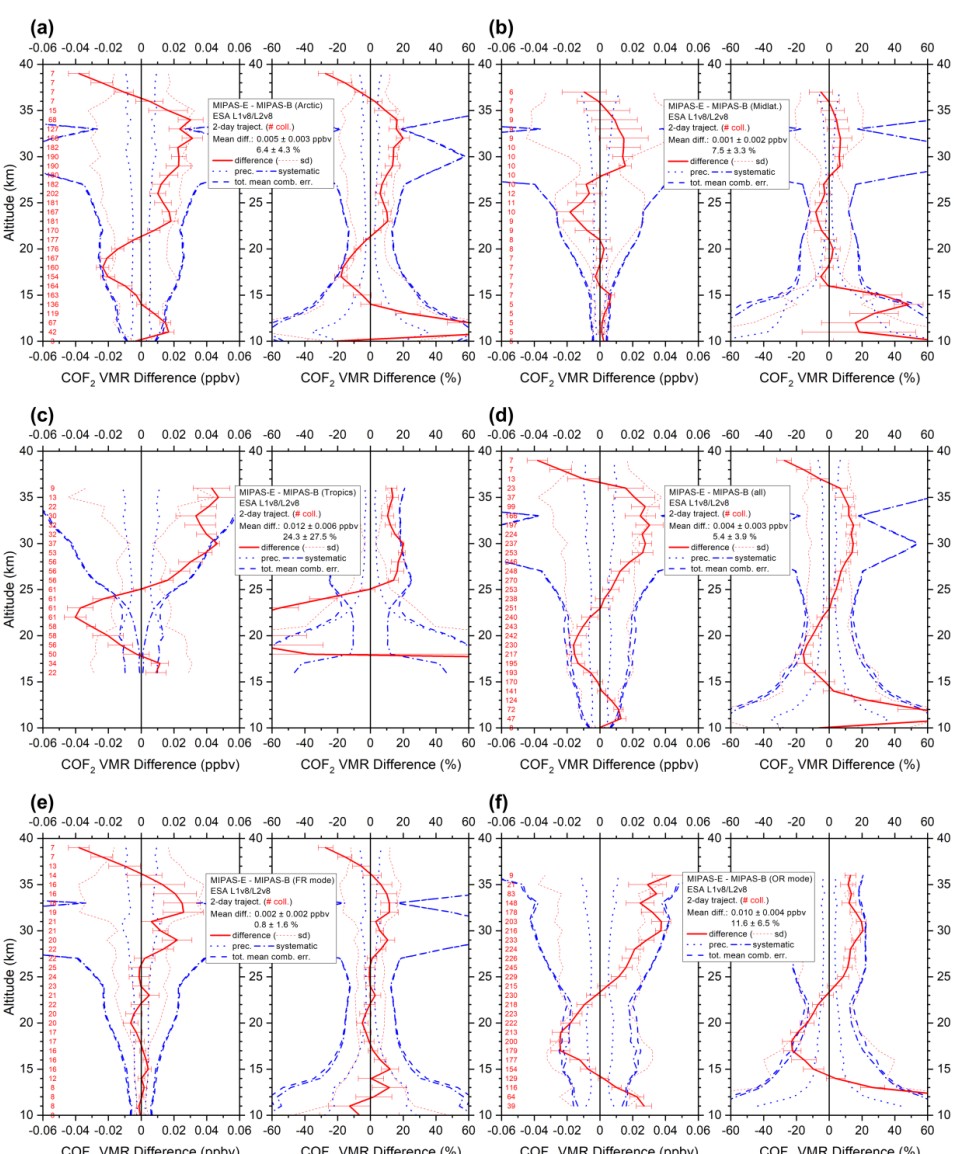

**Figure 16.** Same as Fig. 3 but for $COF_2$.



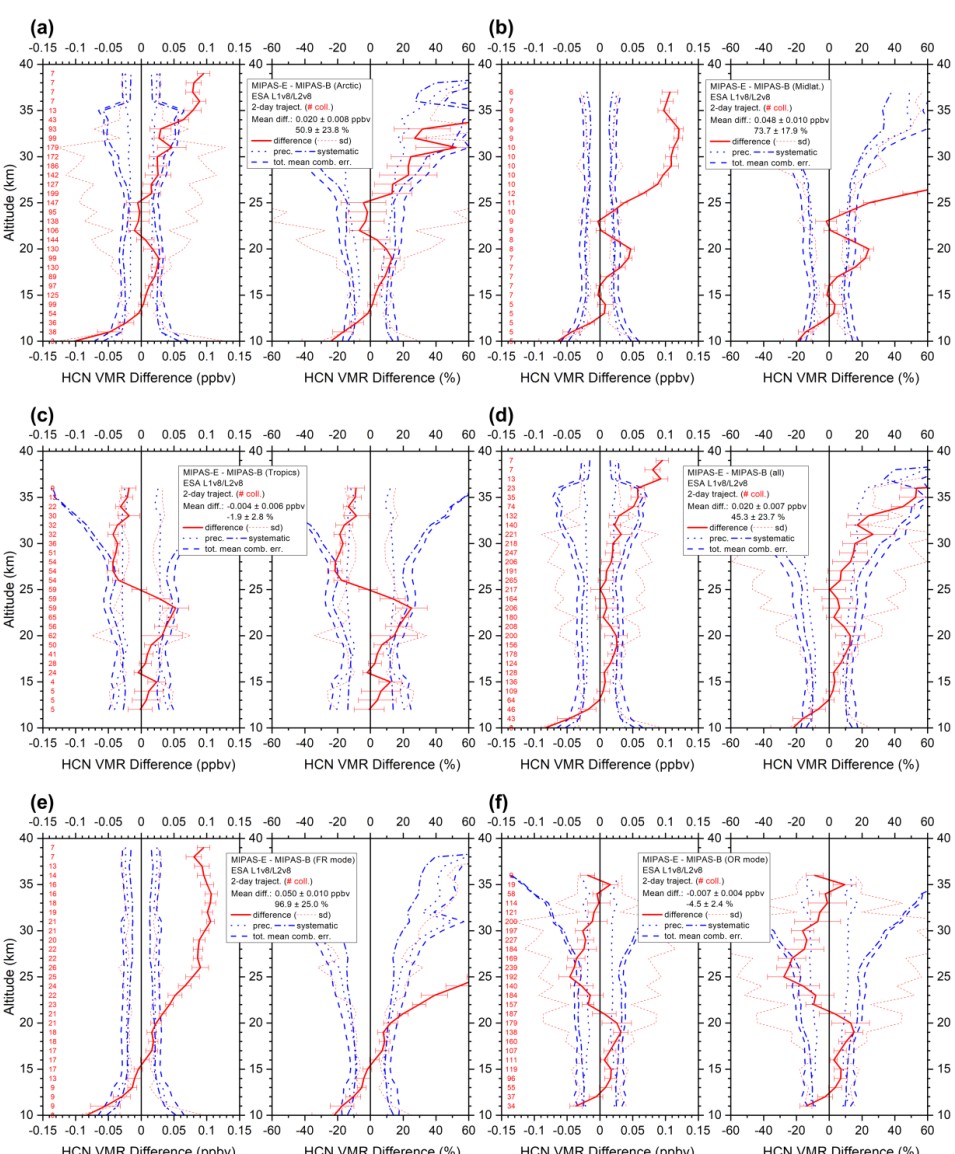

**Figure 17.** Same as Fig. 3 but for HCN.



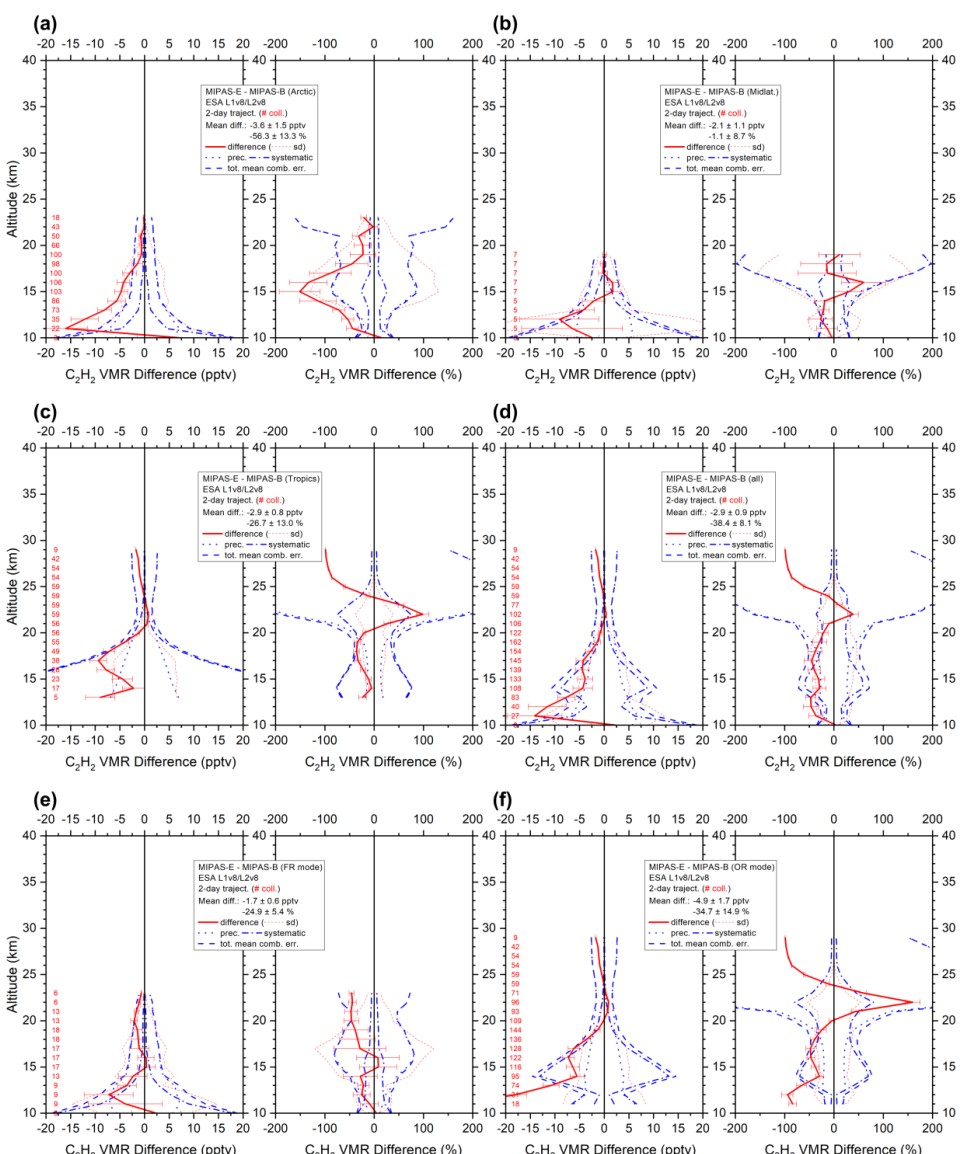

**Figure 18.** Same as Fig. 3 but for $C_2H_2$.



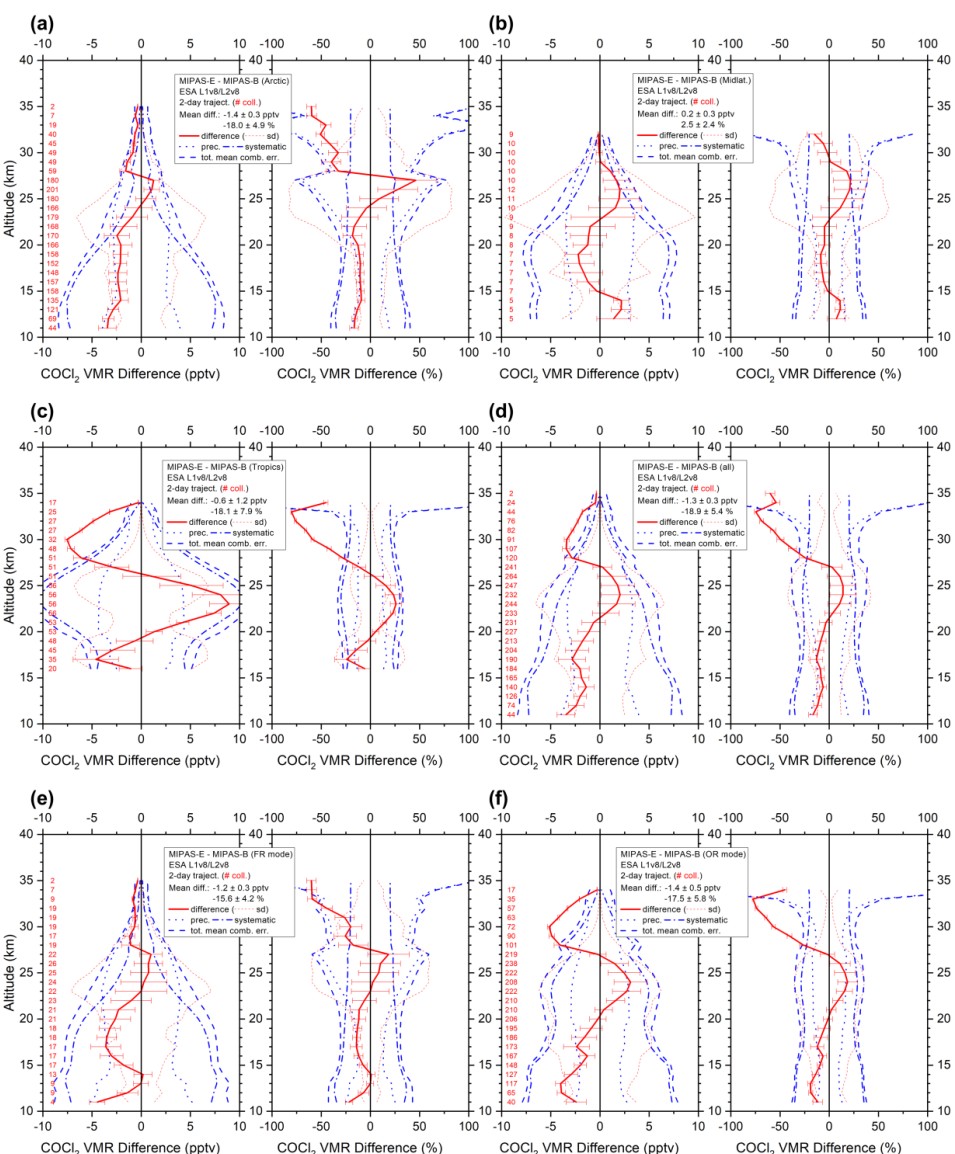

**Figure 20.** Same as Fig. 3 but for $COCl_2$.




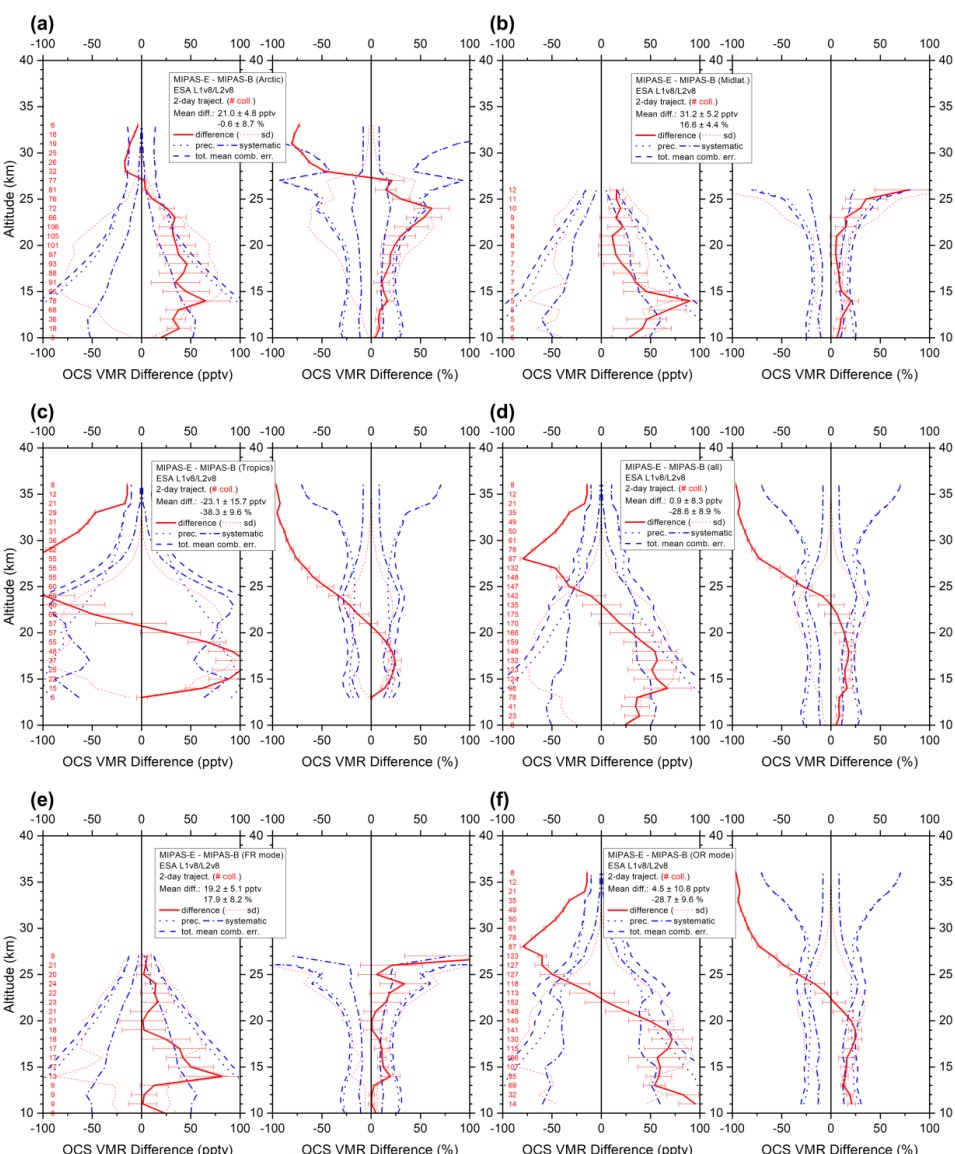

**Figure 21.** Same as Fig. 3 but for OCS.



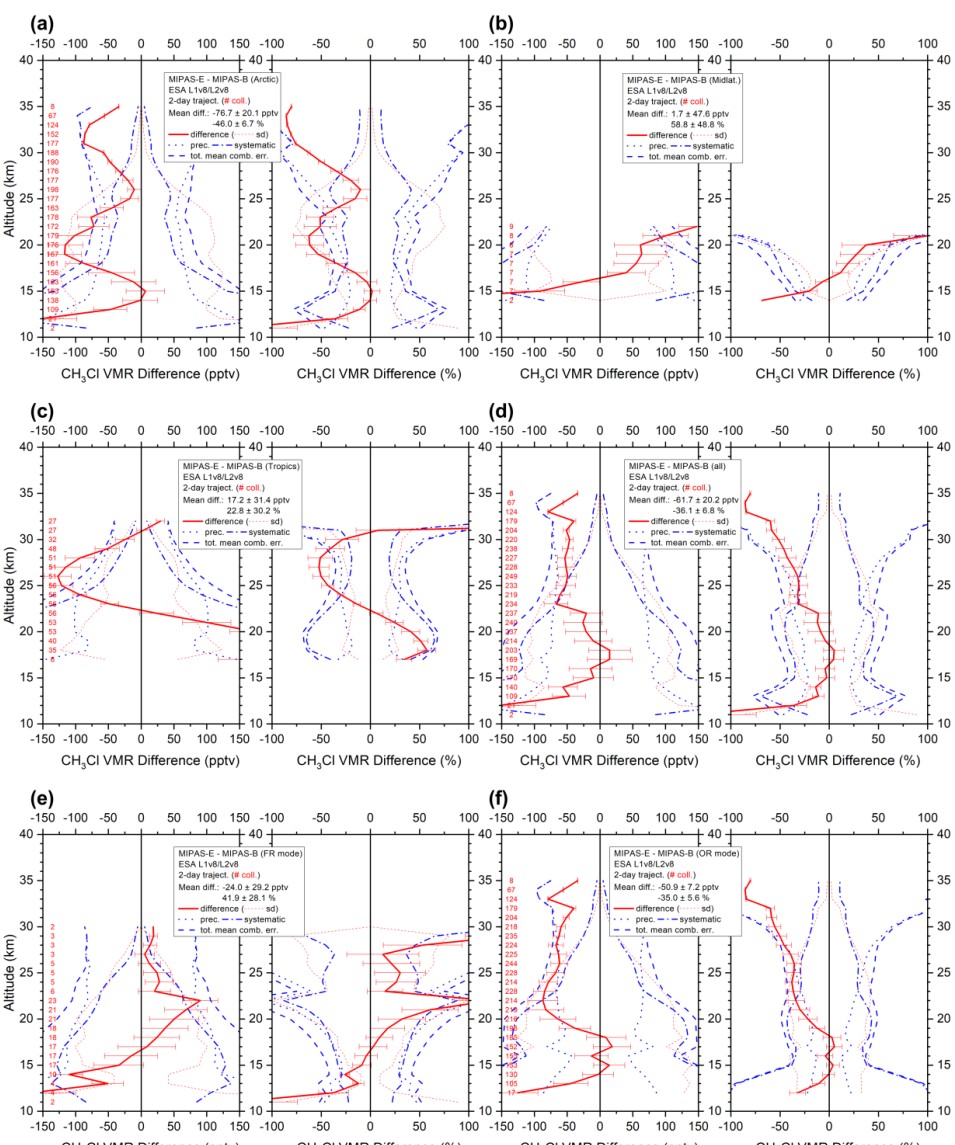

**Figure 22.** Same as Fig. 3 but for CH$_3$Cl.





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
