# Peer review of "Long-term validation of MIPAS ESA operational products using MIPAS-B measurements"

_Atmospheric Measurement Techniques, 2022_

## Author Response (AR1)

**Author's response file**

In the following, we include the answers to the referees. Changes in the text of the revised manuscript are marked with red colour (please see Author's track-changes file).

**Response to Referee 1:**

First of all we thank the referee for the effort to carefully reading the manuscript and for all comments.

**General comments:**

*This manuscript describes a comprehensive validation effort, using balloon-borne profiles from MIPAS-B, to study the biases and associated variability and uncertainties in the retrievals of a large number of VMR profiles from MIPAS aboard ENVISAT (MIPAS-E); some trajectory-based studies are used to provide additional "coincidences" between the balloon and satellite profiles, and more statistical analyses. This work covers the upper troposphere to the mid- to upper stratosphere. While there is no discussion of any systematic temporal changes (or trends), in part because the time period covered (2002-2012) is not quite long enough to study this well enough based on a few balloon flights, there are some noted differences between the two separate time periods when MIPAS-E was observing in different modes (the original full spectral resolution mode, FR, and the post-2004 optimized resolution mode, OR). One of the main conclusions is that the harder to measure species (for both instruments) lead to poorer overall agreement than for the species with stronger signals (and I assume that this is probably not too unexpected).*

*The comparisons are presented in fairly simple ways in a consistent fashion, which makes the large number of plots easier to digest (however, there is an issue with some of the font sizes, see comments later on). The summary Table is a good way to provide top-level conclusions, even if this can be somewhat oversimplified and difficult to do with a broad brush when differences change somewhat rapidly with altitude. Adding some suggested explanations for the larger differences (especially when outside the combined estimated error bars) could be useful, if possible and if not completely speculative. A few more comments regarding other relevant work (in particular, satellite-to-satellite intercomparison results from the SPARC Data Initiative) would be recommended and welcome, as this could reinforce the impression that MIPAS-E might have a real bias (or not), at least in a few specific cases (the same could be done versus ACE-FTS data in particular, since Raspollini et al. have already discussed some of those comparisons, although using a multi-satellite approach as done by the SPARC DI would be viewed as more comprehensive, even if multi-satellite means have potential issues as well, if some measurements are clearly less desirable than others). One should not forget that MIPAS-B is not necessarily "perfect data" either, so untangling a real bias versus just a relative bias can be difficult. It would also help if estimated systematic error bars for the MIPAS-E results were included in one of the Tables, since these values are provided for MIPAS-B, and the combined uncertainties are used (so estimates of error bars for MIPAS-E exist as well). Using a lower to mid-stratospheric range might be good enough for this, or one could consider separating this into two Tables - for two regions where the error bars might be significantly different; I am open to either approach, as long as more information is provided regarding the*

*'typical' satellite error bars (in tabular form). Otherwise, the manuscript is written in a fairly easy to follow manner, and I have no major objections or issues.*

*After a few improvements, which do add up to almost (but not quite) a major revision (see below for more details), I would recommend that this work proceed to publication in AMT, since this topic is well-suited for the AMT Journal (and there is also not much discussion in this manuscript regarding composition changes or processes in the stratosphere, for example). More specific (and also some very minor editorial-type) comments follow.*

Since the points of criticism mentioned here are listed again in the specific comments, we will address them at the appropriate points below.

**Specific comments:**

*L200-201, here, why is a 2-sigma type of criterion not used, namely assign the term "significant difference" only when twice the SEM is smaller than the bias itself?? This is more in agreement with what most scientific studies would argue "significance" applies to (and if you disagree, please give some argument on this topic in your reply and in the text). The main impact might be in the Table of overall conclusions, where you discuss what may be "significant" (or unexplained) differences. Whether many of the plots should be changed is something else to think about - I am not necessarily arguing for this (but please be very specific regarding the meaning of the error bars given in these plots, 1-sigma or two-sigma, it seems that you list and show one sigma results...yes?).*

All error estimations performed in previous MIPAS validation papers (which were cited in the text) refer to the 1-sigma confidence limit. That's why we decided to do the same here (also for reasons of consistency). We added some text to the manuscript here to clarify that all errors refer to the 1-sigma criterion.

*L203, it would not be out of the question that unexplained errors in MIPAS-B could also be invoked to better "explain" relative differences between the two retrievals, at least in some cases possibly; also, unusual atmospheric variability could be partly responsible for a lack of "perfect coincidence" (also, trajectories and associated results are not "perfect" either). I just think that assigning "all" the "blame" for significant (enough) differences to MIPAS-E is not the only possible solution; perhaps you could admit to this without it invalidating the usefulness of MIPAS-B or these results, as I am not suggesting this at all either... "Unexplained relative biases" might well be a more reasonable way of wording this, for example. It would also be useful to mention what the estimated systematic errors for MIPAS-E are, in Table 1, since this could also give the reader some feeling for which retrieval might be expected to be more accurate, if this is sometimes possible to say. However, certain factors like spectroscopic uncertainties (for example) would likely affect both retrievals in the same way, and if this sort of error was a dominant source of error, then neither instrument would be expected to be significantly more accurate (for absolute measurements) than the other... Just showing error bars for MIPAS-B is not really justified, in my view, and since such error bars do exist for MIPAS-E, why not give the reader some feeling for this as well? Are there enough issues in terms of the different satellite retrievals that this becomes a difficult problem to formulate? My issue here is that you have used some estimates, so why not provide at least a first-order example inn Table 1, or a similar Table? Tables do oversimplify things, especially if there is a*

*fair amount of altitude dependence in the estimated uncertainties (error bars), but having something would be better than nothing. Please clarify, in as much as possible.*

As suggested we changed the expression "unexplained biases" to "unexplained relative biases". We also included a new Table 1 which contains typical error bars for MIPAS-E such that these errors are comparable to the ones of MIPAS-B (now Table 2). Of course, MIPAS-B measurements are not the "real truth" (what we do not claim in the manuscript) but we have to mention that special validation sequences were measured during each MIPAS-B flight. The spectral noise was reduced by averaging multiple spectra per elevation angle. Furthermore, the line of sight stabilization is superior. Special care was also taken during the further data analysis to finally get a robust retrieval result. We see from (new) Tables 1 and 2 that MIPAS-B errors are generally somewhat smaller than those of MIPAS-E. Since for tracers like O3 and CH4 differences between both sensors stay mostly within 10%, the influence of unusual atmospheric variability and/or inaccuracies of trajectory calculations appear to be of minor influence.

*L238, it would be good to add just a sentence or so on the main differences between the current manuscript and the Raspollini et al. (2020) document, since these seem to deal with largely the same results. In fact, stating how ACE-FTS comparisons have enhanced these comparisons could be illuminating, especially when the MIPAS-B/MIPAS-E differences are (significantly) larger than one might have expected. On the same topic, I find that you should add at least a few sentences, when appropriate, regarding the results of the SPARC Data Initiative, for some of the species, especially when MIPAS-E biases (with respect to the satellite instrument mean) appear to follow the same tendency that is found here (although it is also interesting if they do not follow this tendency); I realize that the satellite intercomparisons can also be subject to discussion regarding where the "real truth" might be, as it is not necessarily found by showing a multi-instrument climatological mean. Nevertheless, I believe that it is a problem not to mention that document at all (or, actually, the more recent update by Hegglin et al., 2021) and give that work some credit in terms of at least relative bias identification for MIPAS-E; these sorts of studies rely on many more profiles and therefore, in principle, biases can be more robustly identified (although they are also relative biases, and exact knowledge of truth is always a difficult question). On this topic (SPARC DI), I recommend that at least a sentence or so be considered for each of a few of the species mentioned in this manuscript (H2O, N2O, CH4, HNO3, and NO2 are the main ones - while MIPAS-E ozone, in particular, is not seen to have significant issues), if it seems relevant/appropriate - but doing a bit more homework on this issue and adding some additional relevant text would be a change for the better.*

The MIPAS quality readme file (Raspollini et al., 2020) not only includes the validation results related to MIPAS-B, but in addition ground-based, ACE-FTS, lidar, radiosonde, and ozone sonde validation results. We added some text to the corresponding sentence in the manuscript. This quality (documentation) readme file is very comprehensive (177 pages). We had a lot of discussion in the MIPAS Quality Working Group on how we can split all the results into reasonable concise publications. After all, one decision was to publish the intercomparison results between both MIPAS instruments (among other planned validation publications). Hence, this manuscript is not an overall MIPAS validation study of all atmospheric parameters and many instruments but "only" an intercomparison study between two similar instruments. Anyhow, we already included statements concerning the behavior of recognized differences in comparison to previously published peer-reviewed publications (also in the case of ACE-FTS for COCl2). However, we add some more information from the mentioned readme document at appropriate points in the manuscript and we also give comparative information from the

SPARC DI for the gases mentioned above. However, the MIPAS-E data discussed by the SPARC DI was not produced by ESA but with the processor developed and operated by the Institute of Meteorology and Climate Research in cooperation with the Instituto de Astrofísica de Andalucía (von Clarmann et al., AMT, 2, 159-175, 2009).

*L252, it would also be an improvement if you carried out a "gedanken" experiment, in order to at least roughly estimate what altitude uncertainty might be required to lead to such temperature differences (is it 100 m or more than 1 km, say?).*

The altitude uncertainty needed would be about 1 km. In the Tropics, the mean detected tropopause altitude difference between both sensors is up to 1 km. We added this information to the manuscript text.

*L267-270, see the comment for L252 also, is your thinking regarding H2O pure speculation or would there be a reasonable change in altitudes that could account for the observed relative biases in H2O (how large a change in z, if this is something one can explore "on the back of the envelope", without running full retrieval tests?). If this is just pure speculation, it is probably best to remove the text, I would say. If there are changes that can account for both the T and H2O differences, that might start to be more believable.*

As mentioned above in the case of temperature, in the Tropics, the mean detected hygropause altitude difference between both sensors is up to 1 km. We added this information to the manuscript text.

*L289, "in the order of". Also, how do the SPARC DI results compare to these biases in MIPAS-E, i.e. is MIPAS-E on the high side versus other satellite data as well? If so, this might help your argument; if not, it may be more difficult to decide what to conclude – but adding a brief comment on this topic could well be useful.*

Indeed, from the SPARC DI results it is obvious that differences found between MIPAS-B and MIPAS-E in the FR and OR modes also agree with the discrepancies seen in the SPARC DI where MIPAS data were compared to the multi-instrument mean of satellite sensors. We added this message to the text in the corresponding H2O section in the manuscript.

*L300, change "stated" to "mentioned". See my comment above for SPARC DI relevance, please check for CH4 and N2O as well.*

As suggested, we changed "stated" to "mentioned". We checked the SPARC DI relevance and found that positive biases in a comparable order of magnitude are also obvious in the SPARC DI comparison in the stratosphere and upper troposphere, especially in the FR mode period. We added a corresponding sentence to the manuscript text.

*L305, how large is the NO2 photochemical correction compared to the differences (precorrection) between the two data sets? That is, does the correction actually improve the level of agreement? Again, the SPARC DI results might help the interpretation here (worth a try).*

The photochemical correction is in the range of several ppbv above 25 km and actually improves the level of agreement. The SPARC DI differences and associated biases to the multiinstrument mean of satellite sensors show a quite time-variable picture and are therefore difficult to compare with the intercomparison results obtained within this study. However, differences between the MIPAS instruments shown here stay within the standard deviation of the differences revealed in SPARC DI. We added some text in the manuscript.

*Table 3: Please be a little more consistent in terms of the comments when mentioning whether differences are estimated to be significant or not (this gets back to the 1-sigma versus 2-sigma question as well); in particular, there is a mention for H2O regarding differences [generally] being within the combined systematic errors, but why not be more specific also for CH4 and N2O?*

We now mention in the text that the bias is within the combined systematic errors.

*Figure 1: This Figure could have larger fonts for the readers to be able to read the y-axis label (altitude) and the x-axis as well (one can remove the words "Volume Mixing Ratio" and just write "H2O (ppmv)" or "H2O / ppmv"). The larger labels in the plots do allow one to understand which species is shown, but the x-axis and y-axis labels could still be improved; at the same time, this would also allow for a larger font size in the numbers shown along the axes.*

This Figure has been redesigned for larger fonts and captions.

*Figures 2 and similar: I should have mentioned this in the quick review, but the font size for the listed differences in these Figures is probably too small for readers to see well enough on a printed page (without using a zoom feature on the electronic version, even if this is the most likely use of published material these days). It would be good to reduce the unnecessary text and enable larger font sizes for the main comments; also, some things can be abbreviated and some can better be described in the legend(s) instead of inside the various annotated plots.*

In the run-up to the paper, the authors already had some discussions about what the best representation of these figures could be. The current result can be understood as a kind of compromise between different views. However, we emphasize that in the final version the dpi resolution of the images will be clearly improved, so that the images will have more depth of sharpness. Nevertheless, it will not always be avoidable to zoom in on the images to see small details better. Anyhow, since the vast majority of papers are read electronically today, this shouldn't be a big problem. In this respect we have not changed the layout of Figures 2-22.

*L389-390, do the differences between the OR and FR time periods suggest anything regarding the validity of the MIPAS-E data sets (absolute values and scatter or precision)? For example, is the OR mode (in some cases at least) maybe less robust or accurate than the FR mode, or is this too difficult to really ascertain?*

As written in the text, a pronounced bias is visible in the FR phase while no clear bias can be seen in the OR period. The standard deviation between about 20 km and 30 km exceeds the estimated precision in the OR phase. Hence, in the case of HCN the OR period appears to be more reliable compared to the FR phase. However, this message cannot be generalized to all gases.

**Very minor (editorial-type) comments:**

*L24, change "where" to "when".*

Changed.

*L32, I suggest: "This includes an assessment of the data agreement between both sensors, taking into account the combined errors from both instruments."*

Changed.

*L36, "a 5-20% level of agreement between the retrieved... For C2H2,...larger differences (within 20-50%) appear in this altitude range."*

Changed.

*L43, "... operated between 2002 and 2012."*

Changed.

*L52, ...logistical requirement that the satellite...*

Changed.

*L75, solar time of 10:00...*

Changed.

*L77, During each orbit, approximately...*

Changed.

*L89, ...investigations, it was decided...*

Changed.

*L91, back in operation*

Changed.

*L97, "...was steadily increased..." [since this did happen]*

Changed.

*L99-100, "... anomaly occurred, resulting in the loss..."*

Changed.

*L141, comparable to or slightly better than*

Changed.

*L142, overview of*

Changed.

*L150, consistent with*

Changed.

*L155, retrievals [plural might be better here]*

Changed.

*L250, change MIPAS to MIPAS-B for extra clarity.*

Changed.

*L257, add a comma before "we carefully looked at".*

Changed.

*L281, the statistical agreement between the two data sets...*

Changed.

*L321-322, suggesting the need for a more careful use...*

Changed.

*L342, Deviations for CFC-11...are somewhat larger, up to ...*

Changed.

*L346, is also clearly seen if one considers previous comparisons...*

Changed.

*L350, is only available for CCl4 profiles*

Changed.

*L361, positive bias for MIPAS-E...*

Changed.

*L369, which is at the limit of the combined systematic errors.*

Changed.

*L375, There is general agreement between both instruments between ...*

Changed.

*L387, add a comma before "exceeding"*

Changed.

*L393, available for COCl2.*

Changed.

*L406, negative bias in MIPAS-E ...*

Changed.

*L422, acts as a precursor for the stratospheric aerosol layer*

Changed.

*L426, The agreement between the VMR profiles*

Changed.

*L448, a somewhat poorer agreement*

Changed.

*L456, "on the quality of the MIPAS satellite data."*

Changed.

**Response to Referee 3:**

First of all we thank the referee for the effort to carefully reading the manuscript and for all comments.

**General comments:**

*My main concern is how the combined error (Eq. 3) is calculated when comparing the two instruments. One issue is that this seems to neglect any potentially correlated error source between the two instruments. The classic example is a spectroscopic error, where if both retrievals use the same spectroscopic database this equation will overestimate the combined error, but there also may be correlated effects from non-LTE errors or other effects absent in both forward models. I understand that these errors may not be perfectly correlated between the two instruments due to differences in retrieval methods (I see different microwindows were mostly used), spectral resolution, etc., but if the dominant source of estimated systematic error between the two measurements is a potentially correlated error like a spectroscopic error it*

*draws into question some of the conclusions made. It is not easy to correctly account for these correlations, but I would suggest at least stating what the dominant source of systematic error is for each case and analyzing if it is potentially correlated between the two instruments. Since some of the main conclusions of the paper are to recommend caution in areas where the observed differences between MIPAS-E and MIPAS-B is larger than the estimated systematic error it is critical that the estimated systematic error is interpreted correctly.*

As the referee already writes, correct error assessment is a difficult task. However, we did not attribute larger VMR differences between the two MIPAS instruments to spectroscopic inaccuracies of lines when the same spectroscopic parameters for the retrieval of the target gas were used. Nor have we attributed major retrieval discrepancies to other possibly correlated errors. In case where different spectroscopic data for the target gas were used during the retrievals (like $COCl_2$) this was already mentioned in the text. Non-LTE errors do not play a significant role in the MIPAS-B retrievals and are therefore neglected for this instrument. We are sure that we did not treat relevant correlated errors as uncorrelated. "Effects absent in both forward models" as mentioned by the referee are of course not included in the error estimation but such effects actually may lead to "unexplained relative biases" as mentioned in the manuscript text.

*A full validation of every species measured by MIPAS is a monumental task, and the MIPAS-E to MIPAS-B comparisons done by the authors is one piece of that puzzle. This is fine, I don't think the authors need to include more data or analysis, but as a naive MIPAS-E data user some of the results are hard to interpret on their own. The main takeaway that I get is that I should go read the MIPAS product quality document instead (of which a version of this manuscript serves as input to). Once again, this is not a problem by itself, but the manuscript could use some further explanation on how this work fits into the larger body of MIPAS validation efforts.*

The MIPAS product quality document not only includes the validation results related to MIPAS-B, but in addition ground-based, ACE-FTS, lidar, radiosonde, and ozone sonde validation results. We added some text to the corresponding sentence in the manuscript. This quality (documentation) readme file is very comprehensive (177 pages). We had a lot of discussion in the MIPAS Quality Working Group on how we can split all the results into reasonable concise publications. After all, one decision was to make a paper showing the intercomparison results between both MIPAS instruments (among other planned validation publications). Hence, this manuscript is not an overall MIPAS validation study of all atmospheric parameters and many instruments but "only" an intercomparison study between two similar instruments. Anyhow, we already included statements concerning the behaviour of recognized differences in comparison to previously published peer-reviewed publications. However, we add some more information from the mentioned readme document at appropriate points in the manuscript and we also give comparative information from the SPARC Data Initiative for main gases.

**Specific comments:**

*Section 2.1 l. 68: "hereinafter also referred to as MIPAS-E..."*
*There are some places where simply MIPAS (mostly the remainder of this section) is used to refer to MIPAS-E. I would recommend always using MIPAS-E when referring to MIPAS on ENVISAT for clarity.*

We followed the reviewer's suggestion.

*Section 2.1 l. 78: "in steps of 3 km below 45 km"*
*What about above 45 km?*

The steps are coarser above 45 km. We changed the text accordingly to give more information here.

*Section 2.1 l. 94: "... an equivalent improvement in the vertical and horizontal (alongtrack) sampling"*
*I understand details of the sampling for the FR/OR modes of MIPAS can be found elsewhere, but the horizontal/vertical sampling of each mode should be stated in this section. Especially since the change in MOPD is stated.*

We changed the text in the manuscript to give information on the nominal tangent altitudes.

*Section 2.1 l. 118: "All molecules except HDO ..."*
*Is there a fundamental reason why HDO could not be validated as well?*

HDO was not part of the ESA validation contract, since it was not clear at the time the contract was concluded whether HDO could be retrieved from MIPAS-B spectra in a correspondingly high quality.

*Section 2.2:*
*Has there been any validation of the MIPAS-B data products separate from MIPAS-E that could be mentioned here?*

MIPAS-B was also involved in validation of ILAS/ILAS-2, SMILES, GOMOS, and SCIAMACHY measurements. We included corresponding citations in the Introduction part of the manuscript.

*Section 2.2 l. 125: "... MIPAS-B performance is superior, in terms of NESR ..."*
*Is the improvement in NESR from the averaging spectra or is there an instrumental difference that provides better NESR?*

The MIPAS-B NESR for a single spectrum is slightly lower than the one for MIPAS-E. However, the main part of the NESR reduction in the MIPAS-B spectra comes from the multiple averaging.

*Section 2.3 l. 200: "A bias between both instruments is considered significant if the SEM is smaller than the bias itself."*
*Should twice the SEM be used here instead to be at the ~95% confidence interval?*

In principle this is a question of definition. All error estimations performed in previous MIPAS validation papers (which were cited in the text) refer to the 1-sigma confidence limit. That's why we decided to do the same here (also for reasons of consistency). We added a corresponding sentence to the text here to clarify that all errors refer to the 1-sigma criterion.

*Section 2.3 l. 204: "Since the vertical resolution of the atmospheric parameter profiles of both instruments is of comparable magnitude, a smoothing by averaging kernels has not been applied to the observed profiles"*
*I assume that the error estimates for both instruments also do not include the classical ``smoothing error''? If it is then this would cause the error estimates to be inflated since both instruments have a similar vertical resolution.*

Yes, the assumption of the referee is correct. A smoothing error is not contained in the error estimates.

*Section 3 l. 226: "Trajectory matches are based on diabatic 2-day forward and backward trajectories with a collocation criterion of 1 h and 500 km as described in section 2."*
*Is it possible to demonstrate how well the trajectory matching is working? If I understand correctly there are conditions where the measurement locations are collocated enough that trajectory matching is not necessary, maybe this can be used to show the effectiveness of the trajectory matching.*

It is difficult to give an exact number of how well the trajectory calculation works here but 2-day trajectories are nothing special today in terms of accuracy of the analyses (see, e.g. Dee et al., Q. J. R. Meteorol. Soc. 137, 553–597, 2011, and Hoffmann et al., Atmos. Chem. Phys., 19, 3097–3124, 2019). However, we compared direct coincidence VMR profiles between MIPAS-B and MIPAS-E (already applied for several species in the cited older validation papers) and found no significance difference to the trajectory match results. This was also confirmed in a MIPAS temperature validation study by Zhang et al., Journal of Atmospheric and Solar-Terrestrial Physics, 72, 837–847, 2010.

*Section 3.1 l. 245: "... although the standard deviations exceed the expected precision ..."*
*Could this be because the trajectory matching introduces some variance into the comparisons as well? Even if the trajectory matching is perfect the collocation is still only within 500 km and 1 hr, which would contain some atmospheric variability.*

Yes, a certain atmospheric variability can be a part of this effect. However this should be also visible in a similar way in the comparison of the longer-lived tracer species what is not really the case.

*Section 3.1 l. 252: "A possible reason for this difference between both MIPAS sensors could be an inaccuracy in the altitude assignment..."*
*Presumably this error is included in the error budgets of the instruments, you should be able to quantify if this could actually be the case.*

The line of sight error and the connected altitude assignment is included in the error bars. As written in Section 2.3 the primary vertical coordinate of MIPAS-E is pressure whereas for MIPAS-B it is altitude. Hence this inconsistency might cause some inaccuracies due to the interpolation to a common altitude grid. For instance, the tropopause altitude difference between both MIPAS instruments is up to 1 km. This yields also for the hygropause difference. We added this information in the manuscript text.

*Section 3.2 l. 259: "FR and OR mode comparisons show different vertical shapes of the differences between MIPAS-E and MIPAS-B"*

*Is there a significant difference between the retrieved vertical resolution of H2O in the FR and OR modes? Particularly with the strong altitude gradient of H2O a small change in vertical resolution could cause a large observed difference. In general for every species I wonder how much of the difference between the two modes can be explained from the changing averaging kernel.*

Changes in the vertical resolution from the FR mode to the OR phase may cause some different structures in the VMR profile deviations between both MIPAS instruments, especially in the case of strong vertical VMR gradients. We added some text in the manuscript. However, this possible effect is very difficult to quantify also because the MIPAS-B measurement grid was not constant over all flights and the corresponding retrieval averaging kernel has therefore also changed a bit. Furthermore, we made some test calculations in the case of temperature and found that the influence of the varying averaging kernels can virtually always be neglected here (Zhang et al., Journal of Atmospheric and Solar-Terrestrial Physics, 72, 837–847, 2010).

*Section 3.6:*
*Are there any estimates of how much error could be introduced due to an imperfect photochemical correction? I'm wondering if there could be some effect where the balloon flights tend to occur at a similar time each day and so you don't average over an ensemble of random SZA differences, but I'm just throwing things out there.*

The balloon flights were not performed at a similar time each day. The situation of the trajectory matches was also not similar. The number of trajectory matches is dependent on the wind speed (e.g. if there was no wind in the region of the balloon within 2 days we wouldn't get any matches off the balloon). Studies of the simulated ratios of nitrogen compounds like NO2 have shown that chemical models are generally more accurate in terms of (relative) ratios compared to absolute amounts of nitrogen species (see e.g. Wetzel et al., J. Geophys. Res., 107, D16, 10.1029/2001JD000916, 2002). Furthermore, taking the photochemical correction into account improved the agreement between the measured data from both instruments.

**Technical corrections:**

*Section 3.1 l. 250: Is MIPAS here referring to MIPAS-B?*

Yes. We changed this in the text.

**Response to Referee 2:**

First of all we thank the referee for the effort to carefully reading the manuscript and for all comments.

**Major suggestions:**

*Because the authors show results of comparison between MIPAS-E and -B with little discussion on the differences, the title of this paper would be categorized into "Technical note". The authors might be considered this.*

We now have enhanced the discussion on the differences between both MIPAS instruments in the revised version of the manuscript. For main species, we also added a comparison of the differences found here to the MIPAS-E deviations determined as part of the SPARC Data Initiative (which relates to non-operational MIPAS-E data). In addition, we write at least in one sentence for all 20 gases why it is important to measure them. In this respect, we believe that this study represents something more than just a technical note and we have therefore left the title unchanged.

**Minor comments:**

*p.2, L.58, Along with (1) and (2), there is a dynamical approach using potential vorticity and potential temperature field as suggested by Manney et al. (JGR, 2001).*

We added this sentence to the corresponding passage in the manuscript.

*p.5, L.142, Before Table 1, I suggest to add a short comparison among MIPAS-E (FR and OR) and MIPAS-B specifications using a table: vertical resolutions, spectral resolutions, retrieval grids, and so on.*

It is very difficult to give a "short comparison" inside a table. Vertical resolutions are dependent not only on the changing measurement mode of MIPAS-E but also on the species. This holds also for MIPAS-B where the measurement mode is somewhat changing from flight to flight. However, typical retrieval metrics are given in the text. Furthermore, we added a new Table 1 to give more retrieval information for MIPAS-E in analogy to the MIPAS-B retrieval information table.

*p.8, L.218, For SZA corrections, did you change (correct) MIPAS-E data to match the MIPAS-B's SZA only for the daytime?*

The correction factors were calculated for day and nighttime. We added two sentences to make this issue clearer.

**Response to Referee 4:**

First of all we thank the referee for the effort to carefully reading the manuscript and for all comments.

**Detailed comments:**

*Please try to improve the quality of Figures 2-22 by increasing figure resolution, using distinct colors, optimizing line widths, or using shading. For example, blue dashed and dash-dotted lines overlap in many cases. It might look more clear if total error intervals will by indicated by shading.*

In the run-up to the paper, the authors already had some discussions about what the best representation of these figures could be. The current result can be understood as a kind of compromise between different views. However, we emphasize that in the final version the dpi resolution of the images will be clearly improved, so that the images will have more depth of

sharpness. Nevertheless, it will not always be avoidable to zoom in on the images to see small details better. Anyhow, since the vast majority of papers are read electronically today, this shouldn't be a big problem. In this respect we have not changed the layout of Figures 2-22.

*Line 181: "The primary vertical coordinate of MIPAS-E is pressure whereas for MIPAS-B it is altitude." Please specify which temperature/pressure profiles are used for pressure-altitude conversion.*

MIPAS-E pressure altitudes were logarithmically interpolated to the MIPAS-B hydrostatic pressure levels. Hence, vertical profiles refer to the MIPAS-B pressure-altitude grid. This has now been formulated more clearly in the text.

*Lines 182-183: "vertical profiles refer to the MIPAS-B pressure-altitude grid". It is not clear what is used as the vertical coordinate. For "pressure-altitude", please give a formula or a reference.*

This should be clear now from the explanation above. The MIPAS-B altitudes are connected with hydrostatic pressure levels (calculated using the basic hydrostatic equation).

*Lines 200-201: "A bias between both instruments is considered significant if the SEM is smaller than the bias itself." The significance is meaningful only at a specified significance level, which should be specified. Please justify also why you selected a low confidence level (for example, 95% confidence level corresponds to 2-sigma interval).*

All error estimations performed in previous MIPAS validation papers (which were cited in the text) refer to the 1-sigma confidence limit. That's why we decided to do the same here (also for reasons of consistency). We added a corresponding sentence to the text here to clarify that all errors refer to the 1-sigma criterion.

*Line 224: Please define the used threshold for the maximum error.*

This is not one number for all atmospheric parameters. The used thresholds are given in the MIPAS output NetCDF files. We included the reference Raspollini et al. (2022) in the manuscript text, where the quality filtering is described in detail.

*Line 273: reference to WMO ozone assessments would be also appropriate here.*

We added this reference to the text.

*Table 1 is informative and good. An analogous table for MIPAS/Envisat would be useful.*

We prepared a corresponding table and attached it to the text.